# OvA-LP: A Simple and Efficient Framework for Federated Learning on Non-IID Data

## Abstract

Federated fine-tuning (FFT) adapts foundation models to decentralized data but remains fragile under heterogeneous client distributions due to local drift—client-level update divergences that induce systematic bias and amplified variance in the global model. Existing aggregation and personalization approaches largely correct drift post hoc, which can be brittle under extreme Non-IID conditions. We introduce OvA-LP, a minimalist FFT framework that suppresses drift at its source by combining linear probing on a frozen encoder, one-vs-all heads, and a two-stage schedule informed by a bias–variance perspective. OvA-LP demonstrates strong Non-IID robustness and substantially outperforms state-of-the-art PEFT baselines on CIFAR-100 and DomainNet, while maintaining stable performance across participation ratios. Although performance decreases under the most severe domain-shift configuration, OvA-LP exhibits significantly improved stability in practical settings and generalizes across diverse datasets, model architectures, and modalities. These results highlight source-level drift suppression as a viable alternative direction for federated fine-tuning, expanding the design space beyond adaptation-centric approaches.

## 1 Introduction

Foundation models (FMs) have reshaped machine learning by providing powerful pretrained representations that can be adapted to diverse downstream tasks. In federated learning (FL), this shift has led to federated fine-tuning (FFT), where clients adapt a shared encoder rather than training models from scratch (Zhuang et al., 2023). Existing FFT approaches can be grouped into three regimes: full fine-tuning, parameter-efficient fine-tuning (PEFT), and linear probing (LP). Full fine-tuning updates all encoder parameters but is prohibitively expensive and communication-heavy in realistic FL deployments. PEFT freezes the encoder while training lightweight adaptation modules such as adapters, LoRA, or prompts (Houlsby et al., 2019; Hu et al., 2022; Lester et al., 2021), providing adaptation capability at substantially reduced cost and therefore becoming the dominant practical choice in modern FFT pipelines. In contrast, LP updates only a linear classification head and lacks any adaptiveness, which has traditionally led it to be regarded as inadequate for more demanding downstream tasks.

Despite its efficiency and growing adoption, FFT remains highly vulnerable under heterogeneous client distributions (Ren et al., 2025). The central difficulty lies in local drift: distributional differences cause local updates to diverge, and when aggregated they manifest as both systematic bias and amplified variance, degrading accuracy and slowing convergence. In practice, under extreme Non-IID conditions, state-of-the-art FFT approaches often converge slowly and retain far below half of their IID accuracy within a fixed round budget (e.g., 50 rounds), indicating limited ability to control drift at its origin.

Prior work addressing this challenge largely follows two methodological directions. Aggregation strategies modify the global update rule, ranging from FedProx and Scaffold (Li et al., 2020; Karimireddy et al., 2020) to recent variants incorporating LoRA-aware adjustments (Wang et al., 2024; Guo et al., 2024; Yan et al., 2025). Personalization frameworks introduce client-specific components to absorb drift locally, including earlier designs such as FedPer and FedRep (Arivazhagan et al., 2019; Collins et al., 2021) as well as expert- or prompt-based extensions (Hu et al., 2025; Weng et al., 2024). Many contemporary approaches combine these directions; for example, FFT-

MoE integrates PEFT with expert-based personalization, PFPT jointly adapts prompts and aggregation, and FLORA couples LoRA updates with drift-aware optimization((Hu et al., 2025; Weng et al., 2024; Wang et al., 2024)). Despite their variety, existing methods share a common principle: drift is regarded as unavoidable and mitigated only after it emerges. This reactive view leaves systems fragile when heterogeneity becomes severe.

In this work, we present OvA-LP, a minimalist FFT framework that suppresses client drift at its source by preserving pretrained feature geometry rather than adapting it. OvA-LP integrates three lightweight components—linear probing on a frozen encoder, one-vs-all (OvA) heads, and a two-stage training schedule—and unifies them under a bias–variance decomposition of federated gradients. This reframes drift mitigation from post-hoc correction to proactive suppression and suggests a new direction for robustness in federated fine-tuning beyond adaptation-centric FFT pipelines.

Our main findings are:

- OvA-LP effectively suppresses client drift at its origin, providing a principled basis for robust FFT under heterogeneous client distributions.

- Across CIFAR-100 and DomainNet, OvA-LP demonstrates strong non-IID robustness and substantially outperforms PEFT baselines, indicating that the proposed framework is highly effective when domain shift is moderate.

- OvA-LP remains stable across participation ratios and maintains strong performance even at 0.1 participation, indicating practical resilience under realistic federated constraints.

- Extensive evaluations across diverse datasets, model architectures, and modalities exhibit consistent trends, supporting the generality and scalability of the proposed framework.

These results demonstrate that source-level suppression of drift can yield substantial robustness gains under heterogeneous client distributions, and that preserving pretrained representation geometry can serve as a competitive alternative to adaptation when domain shift is moderate. Rather than framing adaptation and preservation as mutually exclusive choices, this perspective broadens the design space for federated fine-tuning and suggests opportunities for complementary strategies that balance the two as needed.

## 2 RELATED WORK

**Aggregation strategies.** A long line of work has sought to improve FL robustness by modifying the global update rule. Classical approaches such as FedProx and Scaffold reduce the variance of client updates and partially stabilize convergence. More recent extensions adapt these ideas to PEFT settings, for example FLoRA, FedSA-LoRA, and FRLoRA (Wang et al., 2024; Guo et al., 2024; Yan et al., 2025). While effective in mitigating some client drift, these methods still rely on aggregation at the server side, typically applied only after local divergence has already occurred.

**Personalization frameworks.** Another direction attaches client-specific modules to absorb drift locally. Examples include FedAdapter and FedPrompt (Cai et al., 2022; Zhao et al., 2023), as well as expert-based extensions such as FFT-MoE and PFPT. These approaches improve local adaptation, but global consistency remains limited because personalization cannot prevent drift from propagating into the shared model.

**Classification heads for label imbalance.** Another line of work modifies the classification head to mitigate skewed label distributions. FedRS (Li & Zhan, 2021) restricts softmax updates for missing classes, mitigating bias under label imbalance. OvA-based approaches such as FedOVA, FedABC, and ATHENA-FL (Zhu et al., 2021; Wang et al., 2023; de Souza et al., 2024) decompose multiclass tasks into binary classifiers to avoid softmax coupling and improve fairness. However, these methods are designed for scratch training and focus mainly on label imbalance, without addressing the broader challenge of feature drift.

**Label noise robustness.** A complementary line of research tackles noisy labels in FL. Methods such as FedCorr (Xu et al., 2022) and FedLTF (Zhan et al., 2025) design correction mechanisms or robust objectives to improve performance under corruption. While effective, these approaches

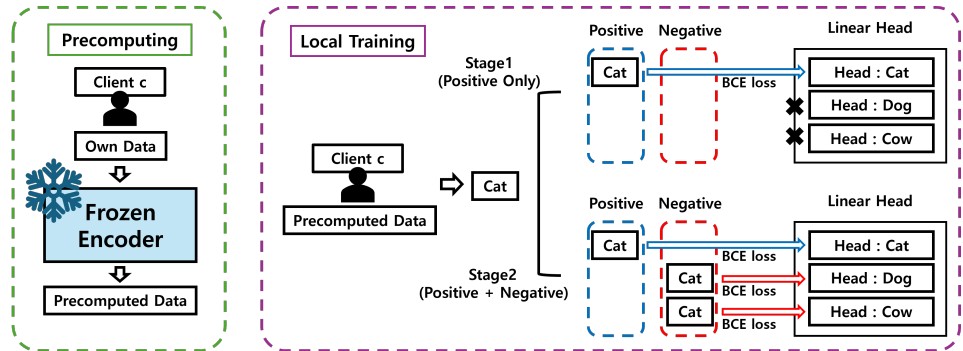

Figure 1: Overall structure of OvA-LP. Clients precompute encoder features once (left) and perform two-stage local training with one-vs-all heads (right).

explicitly address noise rather than the underlying drift mechanisms, and remain orthogonal to our focus.

**Our positioning.** Unlike prior OvA-based methods restricted to scratch training and label imbalance, OvA-LP is designed to prevent drift from arising by freezing the encoder and introducing a two-stage OvA head. Its minimalist design remains modular and in principle compatible with aggregation and personalization families, suggesting potential for deployment across diverse FFT pipelines.

## 3 METHODOLOGY

**Overview.** OvA-LP is motivated by a source-level philosophy: preventing drift at its origin rather than correcting it post hoc. Fig. 1 summarizes the overall workflow. Clients first precompute encoder features with a frozen backbone, then train one-vs-all heads under a lightweight two-stage schedule. This design is guided by a bias–variance decomposition of federated gradients, which identifies local bias, global bias, and variance as the root causes of drift. OvA-LP targets each of these components with simple yet complementary mechanisms: feature geometry bounds the effect of feature skew, OvA heads eliminate label-skew bias and variance amplification, and the two-stage schedule stabilizes optimization under participation variance.

The remainder of this section develops these ideas step by step. Sec. 3.1 formalizes the bias–variance framework that motivates our design. Sec. 3.2 shows how pretrained geometry preserves alignment and separation, limiting bias from feature skew. Sec. 3.3 analyzes label skew, explaining how OvA decoupling removes the bias and variance amplification caused by softmax coupling. Finally, Sec. 3.4 addresses the remaining variance, demonstrating how the two-stage schedule achieves fast and stable convergence. Together, these analyses show how OvA-LP systematically aligns with the bias–variance view to bring Non-IID training close to the IID reference.

### 3.1 BIAS–VARIANCE FRAMEWORK

We begin by formalizing drift through a bias–variance decomposition, which identifies local bias, global bias, and variance as the core sources of degradation.

Client drift under non-IID data can be understood through a bias–variance decomposition at both local and global levels. Let the stochastic gradient on client $i$ be $g_i = \nabla\ell(w; x, y)$. Denote by $\mathcal{D}_i$ the local data distribution of client $i$ and by $\mathcal{D}$ the global distribution. The local loss is $L_i(w) = \mathbb{E}_{(x,y)\sim\mathcal{D}_i}[\ell(w; x, y)]$ with expected gradient $\nabla L_i(w)$, and the global loss is $L(w) = \mathbb{E}_{(x,y)\sim\mathcal{D}}[\ell(w; x, y)]$ with gradient $\nabla L(w)$.

**Local bias.** Each client's optimum deviates from the global one by $b_i = \nabla L_i(w) - \nabla L(w)$, arising from distributional differences across clients, in particular feature skew and label skew.

| Metric | Pretrained (T) | Pretrained (F) |
|---|---|---|
| Alignment $\downarrow$ | $1.366 \pm 0.006$ | $1.761 \pm 0.008$ |
| Intra $\downarrow$ | $0.818 \pm 0.002$ | $0.917 \pm 0.004$ |
| Inter $\uparrow$ | $0.840 \pm 0.004$ | $0.487 \pm 0.008$ |
| Ratio $\downarrow$ | $0.974 \pm 0.007$ | $1.885 \pm 0.038$ |

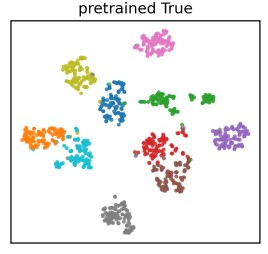

Figure 2: Feature geometry of pretrained vs randomly initialized encoders (CIFAR-10, ViT-L/16).

**Global bias.** Aggregating across clients yields $B = \mathbb{E}_i[\nabla L_i(w)] - \nabla L(w)$, which distorts the overall update direction and accumulates to reduce accuracy.

**Local and global variance.** Even within a single client, stochastic gradients fluctuate with variance $v_i = \text{Var}[g_i]$. When aggregated with weights $p_i$ (e.g., proportional to dataset sizes $n_i$), the update is $\hat{g} = \sum_i p_i g_i$ with variance $V = \text{Var}[\hat{g}]$, which is further amplified by quantity skew.

**Takeaway.** Local bias and variance are the primary contributors to drift, while global bias and variance are their aggregated manifestations. Variance is further exacerbated under label skew due to softmax coupling, which introduces cross-class covariance. This explains why aggregation-level fixes cannot fundamentally solve non-IID degradation: they address only the aggregate symptoms rather than the underlying local causes.

### 3.2 LINEAR PROBING AND FEATURE GEOMETRY

Feature skew is bounded by pretrained geometry: alignment clusters same-class samples, and separation keeps classes apart.

We quantify feature geometry with four standard metrics. Following Wang & Isola (2020), alignment is defined as the expected squared distance between positive pairs:

$$\text{Alignment} = \mathbb{E}_{(x_i, x_j) \sim p_{\text{pos}}} \|f(x_i) - f(x_j)\|_2^2.$$

In addition, we report three well-known statistical measures of representation geometry:

$$\text{Intra} = \mathbb{E}_{x_i \sim y} \|f(x_i) - \mu_y\|_2^2, \qquad \text{Inter} = \mathbb{E}_{y \neq y'} \|\mu_y - \mu_{y'}\|_2^2, \qquad \text{Ratio} = \frac{\text{Intra}}{\text{Inter}}.$$

Here $p_{\text{pos}}$ denotes the distribution over positive sample pairs $(x_i, x_j)$ belonging to the same class, $\mu_y$ is the centroid of class $y$, and $f(\cdot)$ is the encoder representation. Alignment captures the closeness of positive pairs, Intra measures the compactness of each class cluster, Inter quantifies separation between class centroids, and Ratio summarizes the trade-off. Smaller Alignment, Intra, and Ratio and larger Inter indicate stronger feature geometry.

Fig. 2 compares pretrained and randomly initialized encoders. Across all four metrics, pretrained features show smaller Alignment and Intra, larger Inter, and a lower Ratio, confirming that they form compact, well-separated clusters.

From the bias–variance perspective, this structural geometry directly limits the bias induced by feature skew: alignment keeps same-class representations compact, while separation enforces clear boundaries across classes. As a result, client updates remain anchored to the global geometry, and the extent of local bias before aggregation is fundamentally bounded. In the ideal case of perfect alignment, feature-induced bias would vanish entirely.

### 3.3 OVA HEAD AND DECOUPLING

The second major source of drift is label skew, which biases gradients and amplifies variance through softmax coupling. Let $h(x) = f_\theta(x) \in \mathbb{R}^d$ denote the encoder representation of input $x$, and let

$w_c \in \mathbb{R}^d$ be the classifier weight vector for class $c$. We use $1[y = c]$ to denote the indicator for the ground-truth class.

**Softmax coupling.**    For class $c$, the gradient of the cross-entropy loss with respect to $w_c$ is

$$g_c(x, y) = (1[y = c] - p_c(x))h(x), \quad p_c(x) = \frac{\exp(w_c^\top h(x))}{\sum_{j=1}^{K} \exp(w_j^\top h(x))}.$$

Because all classes share a denominator, majority classes repeatedly dominate updates, while minority classes receive little signal. As analyzed in FedRS (Li & Zhan, 2021), this coupling introduces both bias and variance amplification under label skew. Replacing softmax with independent OvA heads removes this cross-class covariance, eliminating the mechanism behind label-skew drift. As analyzed in FedRS (Li & Zhan, 2021), majority classes dominate through repeated "pulls," while minority classes often receive only "pushes." This imbalance introduces bias, since updates are driven by probability-weighted terms $p_c(x)$ rather than purely class-specific targets. It also amplifies variance, because the shared denominator induces non-zero cross-class covariances $\mathrm{Cov}(g_c, g_j) \neq 0$. Together, these mechanisms destabilize training under heterogeneous distributions.

**OvA decoupling.**    An OvA head replaces softmax with independent binary classifiers. The gradient of the logistic loss with respect to $w_c$ is

$$g_c^{\mathrm{OvA}}(x, y) = (1[y = c] - q_c(x))h(x), \quad q_c(x) = \sigma(w_c^\top h(x)) = \frac{1}{1 + \exp(-w_c^\top h(x))}.$$

$q_c(x)$ is the Bernoulli likelihood under a logistic regression head, and each head optimizes its binary logistic loss independently. As a result, the pull/push imbalance described in FedRS disappears: majority and minority classes are updated without mutual interference. This decoupling eliminates the mechanism of label-skew-induced bias and variance amplification, directly addressing the sources of drift at their origin.

### 3.4    VARIANCE AND TWO-STAGE TRAINING

After bias terms are suppressed, variance remains the main source of drift. Variance cannot be eliminated entirely, but its destabilizing effect can be controlled through a two-stage curriculum aligned with the OvA structure.

**Stage 1 (positive-only).**    When pretrained representations preserve alignment and separation, the global optimum of each OvA head lies near the class centroid at the point of maximum margin. Training only on positives thus pulls classifier weights toward these centroids, leading to rapid convergence without cross-class conflicts and helping to overcome the destabilizing effect of variance in the early rounds.

**Stage 2 (positive+negative).**    After centroids are established, a large set of negatives is introduced to expand inter-class margins. At the same time, a small fraction of positives is retained as anchors, preventing the decision boundary from drifting away under the stronger influence of negatives. This combination enables efficient margin learning while preserving the stability achieved in Stage 1.

**Takeaway.**    Together, the two stages implement an easy-first, hard-later curriculum. Stage 1 quickly aligns classifiers with class centroids under minimal variance, while Stage 2 leverages negatives for margin expansion without destabilizing the positive clusters. This design directly overcomes variance effects at their source, complementing OvA-LP's treatment of feature and label skew.

## 4    EXPERIMENTS

### 4.1    EXPERIMENTAL SETUP

**Shared setting.**    Our primary experiments use CIFAR-100 with 100 clients for 50 rounds, a scale larger than those adopted in recent FL benchmarks (see Appendix B.1 for survey). We fix five random seeds (0, 42, 777, 1337, 15254) across all runs for comparability. For the IID setting, data

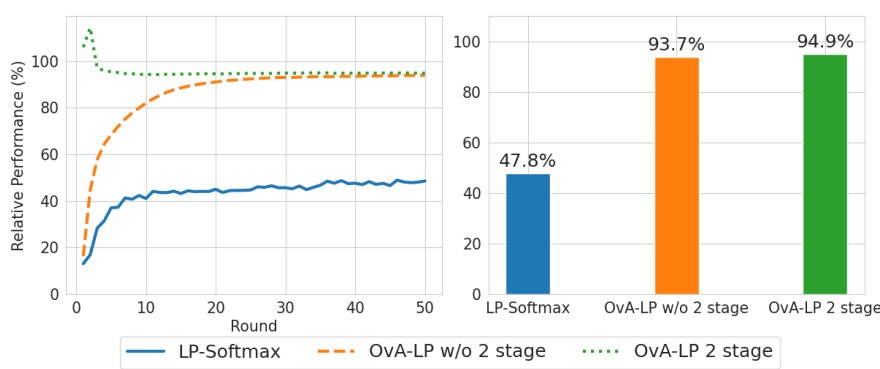

Figure 3: Ablation of OvA-LP components. Stepwise gains ($47.8 \rightarrow 93.7 \rightarrow 94.9$) illustrate the effects of OvA decoupling and two-stage training.

are split uniformly at random across clients. For the Non-IID setting, we follow the FedCorr construction (Xu et al., 2022) and employ a Dirichlet-based partitioning ($p = 0.1, \alpha = 0.001$),widely adopted in the literature.

**Our model.** OvA-LP uses a frozen ViT-L/16 encoder and is trained with 100% client participation, three local epochs per round, batch size 50, learning rate 0.01, and AdamW optimizer with weight decay $1 \times 10^{-4}$. Client updates are aggregated by FedAvg (McMahan et al., 2017), with the first round conducted using Stage 1 (positive-only) training and all subsequent rounds using Stage 2 (positive+negative) training, as described in Sec.3.4.

**Evaluation philosophy.** We quantify non-IID robustness by comparing accuracy trajectories to the IID reference. For round $t$, we compute the relative ratio $R(t) = \text{Acc}_{\text{NonIID}}(t)/\text{Acc}_{\text{IID}}(t) \times 100$. We present results in two unified views: round-wise $R(t)$ curves showing how quickly and stably each method tracks the IID trajectory, and final $R(50)$ barplots summarizing the endpoint gap for each partition. This framing provides a consistent lens that captures convergence speed, stability, and final accuracy.

## 4.2 ABLATION STUDY

We examine the contribution of each design component of OvA-LP by comparing three head configurations: (i) LP with softmax, (ii) OvA-LP without the two-stage design, and (iii) the full OvA-LP.

As shown in Fig. 3, the progression is stepwise. LP-softmax reaches only 47.8% under Non-IID, reflecting limited benefit from encoder freezing alone. Replacing the softmax with independent OvA classifiers stabilizes training and raises performance to 93.7%. Adding the two-stage design enables faster convergence to 94.9%, closely tracking the IID curve within only a few rounds.

A brief overshoot above the IID curve occurs in the first few rounds. This behavior arises from FedAvg's weighted averaging under imbalanced partitions and quickly settles.

In summary, OvA decoupling mitigates label-skew effects, and the two-stage procedure helps overcome variance, leading to faster and more stable convergence. These observations align with the bias–variance decomposition described in Sec. 3.4. Full accuracy curves and efficiency breakdowns are reported in Appendix A.2.

## 4.3 COMPARISON WITH BASELINES

We compare OvA-LP against recent state-of-the-art FFT baselines: PFPT, FFT-MoE, and FLoRA, which employ PEFT-based prompt tuning, mixture-of-experts routing, and LoRA-based adaptation, respectively. LP-softmax is also included as a standard linear probing baseline. Fig. 4 summarizes relative performance under participation ratios 0.1 and 1.0. Full accuracy curves and efficiency breakdowns are reported in Appendix A.3.

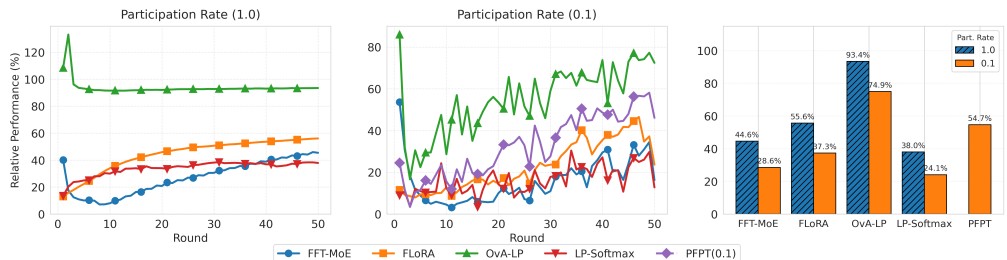

Figure 4: Comparison with baselines under participation rates 0.1 and 1.0. OvA-LP achieves the highest performance in both settings (74.9% and 93.4%). PFPT with full participation (1.0) is omitted due to impractical runtime, which grows nonlinearly ($34s \rightarrow 3,754s$ per round; 100 hours per seed).

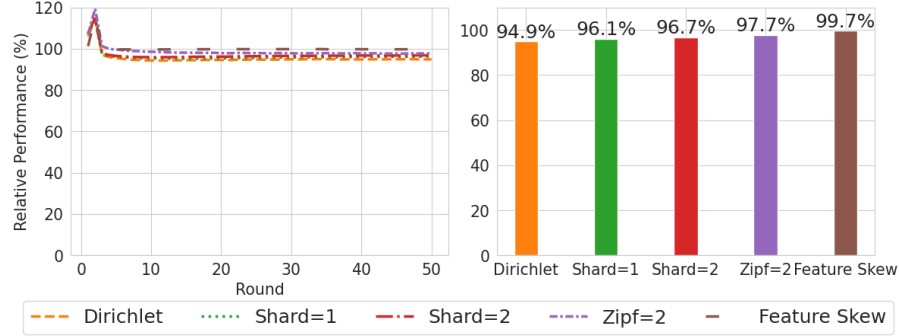

Figure 5: Partition-wise robustness across five heterogeneity patterns. OvA-LP converges to 94.9%–99.7% of IID accuracy.

We use the shared setting described in Section 4.1, and to ensure fair comparison and isolate the effect of adaptation strategies, we unify the core configuration across all methods: all models use a ViT-B/32 encoder and batch size 64, while remaining hyperparameters follow each original implementation (Appendix B.2). This removes confounding effects due to architectural or batch-size differences and enables a clean assessment of Non-IID robustness.

At full participation (1.0), OvA-LP reaches near-IID performance (93.4%), demonstrating effective source-level drift suppression when full aggregation is available. Under partial participation (0.1), accuracy naturally declines due to increased variance; however, OvA-LP still outperforms all baselines by approximately 20–50%, demonstrating strong stability even when only a small fraction of clients participates per round.

In contrast, LP-softmax shows the weakest robustness (24.1% and 38.0%), despite reducing feature–skew–induced bias through representation preservation. The substantial gap between LP-softmax and OvA-LP highlights that linear probing alone is insufficient and illustrates the synergistic effect of OvA heads and the two-stage procedure within our bias–variance framework—indicating that robustness arises from coordinated source-level mechanisms rather than isolated components.

Finally, CIFAR-100 exhibits minimal domain shift relative to ViT pretraining, making this a scenario inherently favorable to adaptation-free approaches. Additional analyses on domain-shifted tasks (DomainNet, EMNIST, AGNews) are presented in Section 5.

## 4.4 ADDITIONAL ANALYSES

We further evaluate the generality of OvA-LP under broader settings beyond the shared FFT setup.

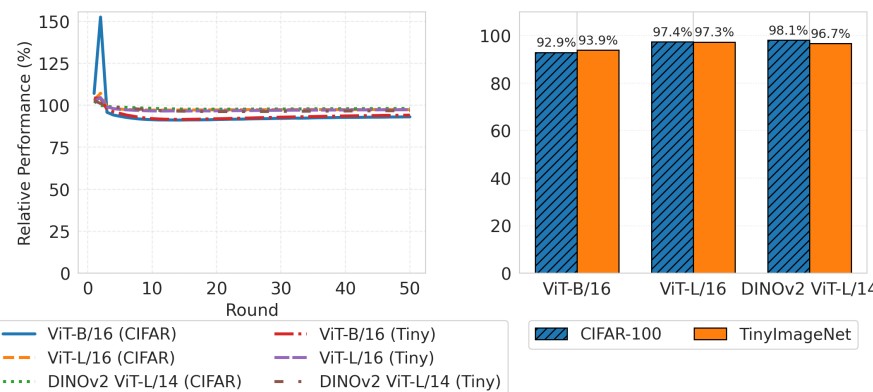

Figure 6: Encoder and task variations. OvA-LP is evaluated with different encoders (ViT-B/16, ViT-L/16, DINOv2-L/14) on CIFAR-100 and extended to TinyImageNet under Dirichlet partitioning.

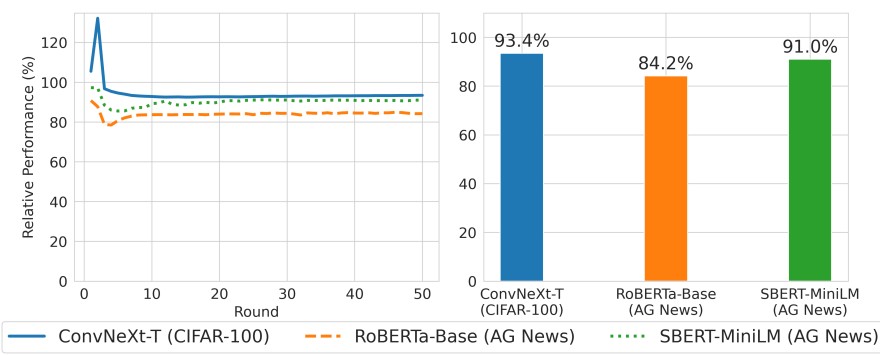

Figure 7: Relative performance across backbone architectures. All models retain 84.2%–93.4% of IID accuracy, demonstrating consistent robustness across both vision and NLP encoders.

### 4.4.1 PARTITION-WISE ROBUSTNESS.

We evaluate OvA-LP under five representative non-IID configurations. In addition to the standard Dirichlet benchmark ($\alpha = 0.001, p = 0.1$), we test four complementary settings: Zipf quantity skew ($s = 2.0$) (Piantadosi, 2014), feature-based clustering via $k$-means, Shard-1 (one class per client), and Shard-2 (two classes per client).

Fig. 5 shows that across all five settings, accuracy trajectories $R(t)$ closely follow the IID curve and final accuracy $R(50)$ ranges from 94.9% to 99.7%. These results demonstrate robustness across diverse non-IID forms, including distributional, quantity, and feature-level skew.

### 4.4.2 ENCODER VARIATION ACROSS DATASETS.

We next test whether robustness depends on encoder scale or task domain. Fig. 6 compares ViT-B/16, ViT-L/16, and DINOv2-L/14 on CIFAR-100, and extends to TinyImageNet under Dirichlet partitioning. Absolute accuracy decreases for the smallest encoder (ViT-B/16), while ViT-L/16 and DINOv2-L/14 remain comparable, with minor fluctuations depending on the dataset. Crucially, in all cases $R(t)$ curves consistently track the IID trajectory, confirming that OvA-LP's robustness is agnostic to encoder scale, architecture, and task domain.

We note that the brief overshoot observed in the first round is a benign effect of size-weighted FedAvg, which appears more prominently with smaller encoders. It stabilizes quickly and does not affect final convergence.

| Noise Type | Symmetric | | | | | Asymmetric | | |
|---|---|---|---|---|---|---|---|---|
| Method | Baseline Acc (%) | | Decline Rate(%) ↓ | | | Baseline Acc (%) | Decline Rate(%) ↓ | |
| Noise Ratio | 0.30 | 0.40 | 0.50 | 0.60 | 0.70 | 0.20 | 0.30 | 0.40 |
| FedAvg | 16.75% | 15.70% | 23.70% | 37.49% | 51.46% | 18.85% | 13.37% | 30.93% |
| Symmetric CE | 16.99% | 17.77% | 25.66% | 40.79% | 49.79% | 26.14% | 17.71% | 36.34% |
| Co-teaching | 34.21% | 8.68% | 36.07% | 51.04% | 66.99% | 34.19% | 20.10% | 33.23% |
| FedCorr | 32.15% | 13.50% | 26.59% | 41.65% | 62.64% | 41.12% | 13.47% | 30.81% |
| FedNoRo | 38.58% | 9.46% | 19.57% | 35.38% | 43.36% | 45.42% | 9.82% | 26.97% |
| FedLTF (Stage 2) | 55.23% | 3.73% | 8.73% | 12.18% | 21.24% | 52.63% | 10.94% | 26.51% |
| FedLTF (Stage 3) | 58.43% | 3.70% | 9.24% | 14.65% | 20.91% | 57.78% | 8.71% | 24.80% |
| OvA-LP (Ours) | 88.78% | **0.76%** | **2.35%** | **4.52%** | **10.35%** | 89.28% | **0.63%** | **1.53%** |

Table 1: Label-noise robustness on CIFAR-100 measured as accuracy decline (%). Baseline results are taken from FedLTF (Zhan et al., 2025) (Table 2). OvA-LP exhibits the smallest degradation, surpassing FedLTF and prior methods.

| Methodology | | IID | | | | | Non-IID | | | | |
|---|---|---|---|---|---|---|---|---|---|---|---|
| Dataset | Part. Rate | LP | OvA | FLoRA | PFPT | FFT-MoE | LP | OvA | FLoRA | PFPT | FFT-MoE |
| cifar-100 | 0.1 | 86.18 | 87.18 | **90.02** | 88.44 | 89.66 | 11.0 | **63.14** | 21.3 | 40.78 | 14.59 |
| | 1.0 | 87.64 | 87.73 | **90.49** | - | 89.81 | 33.22 | **82.0** | 50.73 | - | 40.81 |
| domainnet (clipart) | 0.1 | 78.88 | **85.42** | 66.8 | 69.47 | 60.62 | 10.09 | **58.12** | c | 32.46 | c |
| | 1.0 | 83.14 | **88.16** | 72.42 | 69.13 | 61.09 | 10.94 | **68.1** | 16.69 | - | c |
| domainnet (painting) | 0.1 | 77.19 | **81.5** | 68.22 | 69.13 | 63.51 | c | **54.92** | 14.64 | 36.15 | c |
| | 1.0 | 79.55 | **83.96** | 71.07 | - | 63.6 | c | **65.14** | 27.36 | - | 18.02 |
| eminst | 0.1 | 82.23 | 81.99 | **86.94** | 82.82 | 86.22 | c | 10.17 | c | **16.78** | 14.27 |
| | 1.0 | 83.27 | 82.86 | **86.77** | - | 86.47 | 20.48 | 48.16 | **54.52** | - | 26.34 |

Table 2: Comparison of different methodologies across datasets and participation rates. Reported values are actual accuracy (%), and c indicates model collapse (accuracy < 10%).

### 4.4.3 Cross-Architecture and Task Generalization.

We additionally evaluate OvA-LP on ConvNeXt-Base (CIFAR-100) and RoBERTa-Base / SBERT-MiniLM (AG News), achieving 84.2%–93.4% relative performance (Fig. 7). This confirms that source-level drift suppression extends beyond ViT models to CNN and NLP tasks.

### 4.4.4 Label Noise Robustness.

Following FedLTF (Zhan et al., 2025), we adopt the same label noise benchmarks and directly compare against the baselines it reports. FedLTF represented the prior state-of-the-art under label corruption. As Table 1 shows, OvA-LP achieves markedly smaller accuracy declines, surpassing FedLTF and all other reported methods.

## 5 Discussions

**Domain-shift experiments.** Table 2 summarizes results across datasets exhibiting different levels of domain shift, using a shared experimental configuration (Section 4.3). CIFAR-100 represents a standard natural-image classification task, DomainNet-Clipart and DomainNet-Painting introduce substantial stylistic variation, and EMNIST constitutes an extreme setting in which handwritten symbol structure diverges sharply from natural images.

A common expectation is that LP-based approaches should struggle under domain shift, whereas PEFT methods should benefit from encoder adaptiveness. However, Table 2 shows that LP-Softmax and OvA-LP achieve slightly lower accuracy on EMNIST but substantially higher accuracy on DomainNet compared to FLoRA, PFPT, and FFT-MoE, even under IID conditions—despite the absence of encoder updates. Since IID settings contain neither client drift nor participation-induced variance, these results suggest that modern pretrained encoders such as ViT may already provide feature spaces that are sufficiently linearly separable for typical downstream tasks, reducing the practical necessity of adaptation in such settings.

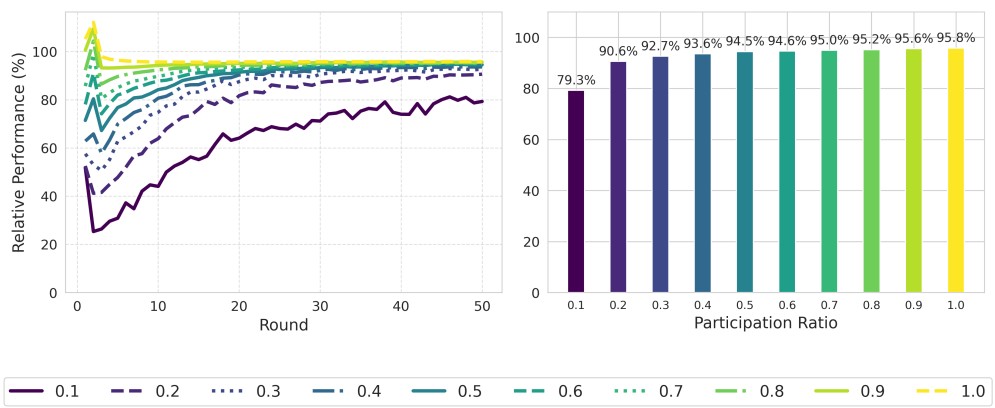

Figure 8: Impact of participation ratio on Non-IID robustness. OvA-LP converges more slowly as participation decreases yet preserves stable performance across all settings.

**Non-IID behavior.** Under Non-IID conditions, performance differences become more pronounced. As shown in Table 2, PEFT approaches degrade substantially—and in some cases collapse (denoted "c")—when exposed to heterogeneous client distributions, aligning with reports that local adaptation can interact adversely with heterogeneity by amplifying gradient variance. Although domain shift still induces larger decreases than in CIFAR-100, OvA-LP demonstrates the strongest performance in most settings, with the exception of the most extreme domain-shift scenario represented by EMNIST.

**Participation ratio.** Figure 8 reports results obtained by varying only the client participation ratio while holding all other settings fixed (Section 4.1). OvA-LP performance decreases as participation is reduced, reflecting increased variance due to limited client visibility. At the minimum participation level (0.1), OvA-LP still exceeds the performance of competing methods on DomainNet and CIFAR-100, indicating robustness under practical FL constraints. In contrast, on EMNIST, OvA-LP does not outperform PEFT methods, and the result at 0.1 participation (10.17%) represents the worst case observed in our experiments. This setting combines the strongest domain shift and the highest participation-induced variance, highlighting a natural limitation of the approach and suggesting directions for future work on controlled or selective adaptation.

## 6    CONCLUSION

We introduced OvA-LP, a minimalist federated fine-tuning framework that suppresses client drift at its source by combining linear probing on a frozen encoder, one-vs-all heads, and a two-stage training schedule. Guided by a bias–variance perspective, OvA-LP adopts a proactive stance toward robustness—preventing drift before aggregation rather than correcting it post hoc.

Experiments on CIFAR-100, DomainNet, and EMNIST show that OvA-LP provides strong performance across both IID and Non-IID settings, and offers substantially improved stability under heterogeneous client distributions. While OvA-LP demonstrates clear advantages on standard and moderately shifted visual domains, results also reveal a limitation in the most extreme domain-shift scenario represented by EMNIST, where PEFT-based approaches remain beneficial. These findings suggest that the practical value of encoder adaptation depends on the severity of domain shift, and that relying solely on adaptation is not universally advantageous even in federated fine-tuning.

This observation offers a new perspective on federated fine-tuning, which has been predominantly driven by PEFT-based approaches, suggesting that source-level control may serve as an alternative path toward robustness. OvA-LP establishes a strong practical baseline under realistic round budgets and highlights the potential of proactively managing drift at the source. A promising direction for future research is exploring hybrid strategies that integrate controlled adaptation with source-level stability, particularly under severe domain shift or limited participation.

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

## A  ADDITIONAL EXPERIMENTAL RESULTS

### A.1  MEASUREMENT DEFINITION.

Time and communication metrics are reported per client, assuming parallel client execution. Total time is computed as the sum of client and server computation per round multiplied by the number of rounds required to reach 95% accuracy (Acc@95). Total communication represents per-client upload traffic per round multiplied by the number of rounds to convergence. Server-side broadcast is omitted because it is identical across all methods and does not affect relative comparison.

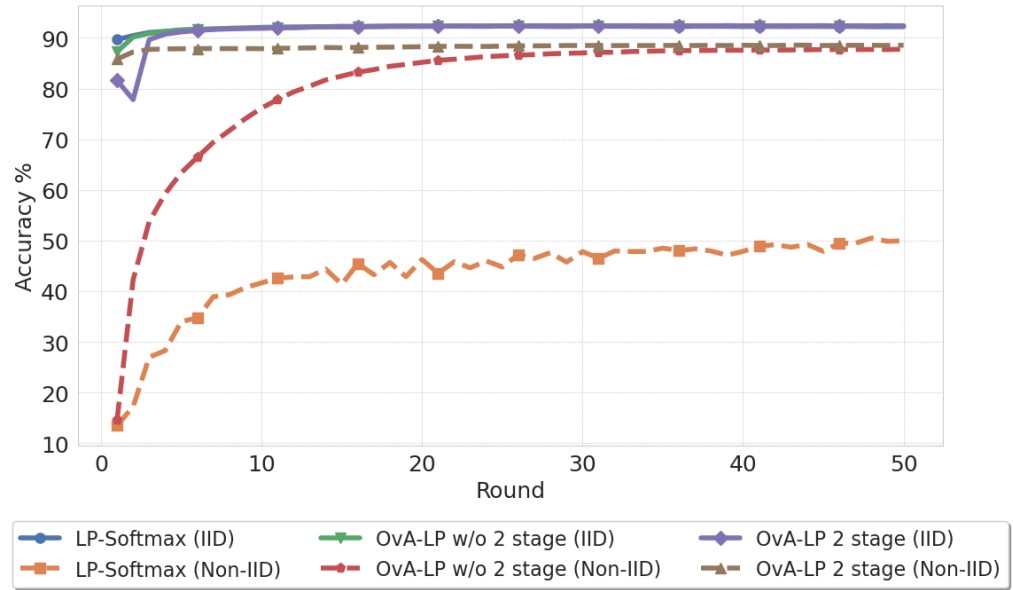

Figure 9: Ablation curves under IID and averaged Non-IID settings. Accuracy trajectories over 50 rounds.

| Methodology | Accuracy (%) | Acc@95 (Rounds) | Total Time (s) | Total Comm. (MB) |
|---|---|---|---|---|
| LP-Softmax (IID) | $92.40 \pm 0.05$ | $1 \pm 0$ | $0.02 \pm 0.00$ | $0.39 \pm 0.00$ |
| LP-Softmax (Non-IID) | $51.52 \pm 2.45$ | $29 \pm 12$ | $0.45 \pm 0.18$ | $11.18 \pm 4.55$ |
| OvA-LP w/o 2 stage (IID) | $92.48 \pm 0.06$ | $2 \pm 0$ | $0.03 \pm 0.00$ | $0.78 \pm 0.00$ |
| OvA-LP w/o 2 stage (Non-IID) | $87.80 \pm 0.31$ | $17 \pm 1$ | $0.25 \pm 0.01$ | $6.49 \pm 0.35$ |
| OvA-LP 2 stage (IID) | $92.47 \pm 0.09$ | $3 \pm 0$ | $0.04 \pm 0.00$ | $1.17 \pm 0.00$ |
| OvA-LP 2 stage (Non-IID) | $88.64 \pm 0.30$ | $1 \pm 0$ | $0.02 \pm 0.00$ | $0.39 \pm 0.00$ |

Table 3: Final performance metrics of ablation study, including accuracy, convergence rounds (Acc@95), total time, and total communication until convergence.

| Methodology | Time (Client, ms) | Time (Server, ms) | Comm. (Client) | Comm. (Server) |
|---|---|---|---|---|
| LP-Softmax (IID) | $13.52 \pm 0.09$ | $1.62 \pm 0.01$ | 400.39 KB | 39.10 MB |
| LP-Softmax (Non-IID) | $14.11 \pm 0.11$ | $1.64 \pm 0.03$ | 400.39 KB | 39.10 MB |
| OvA-LP w/o 2 stage (IID) | $12.74 \pm 0.11$ | $1.61 \pm 0.02$ | 400.39 KB | 39.10 MB |
| OvA-LP w/o 2 stage (Non-IID) | $13.26 \pm 0.18$ | $1.62 \pm 0.02$ | 400.39 KB | 39.10 MB |
| OvA-LP 2 stage (IID) | $12.63 \pm 0.08$ | $1.66 \pm 0.03$ | 400.39 KB | 39.10 MB |
| OvA-LP 2 stage (Non-IID) | $13.39 \pm 0.16$ | $1.64 \pm 0.02$ | 400.39 KB | 39.10 MB |

Table 4: Per-round computation and communication costs of ablation study.

## A.2 ABLATION RESULTS

Fig. 9 and Tables 3, 4 report ablation results on CIFAR-100 with 100% participation. Time and communication metrics are computed per client assuming parallel execution (Appendix A.1).

Under IID, all variants converge to similar accuracy. Under Non-IID, LP-Softmax requires 29 rounds (0.45 s, 11.18 MB) to reach Acc@95, OvA-LP w/o two-stage requires 17 rounds (0.25 s, 6.49 MB), and full OvA-LP converges in a single round (0.02 s, 0.39 MB).

Per-round cost remains effectively identical to LP-Softmax (13.39 ms client, 1.64 ms server, 400.39 KB upload). Thus, OvA-LP matches the cost profile of the lightest FFT baseline while

accelerating convergence, enabling the striking result of 0.02 s total training time and 0.39 MB communication in this ideal setting.

## A.3 BASELINE COMPARISONS

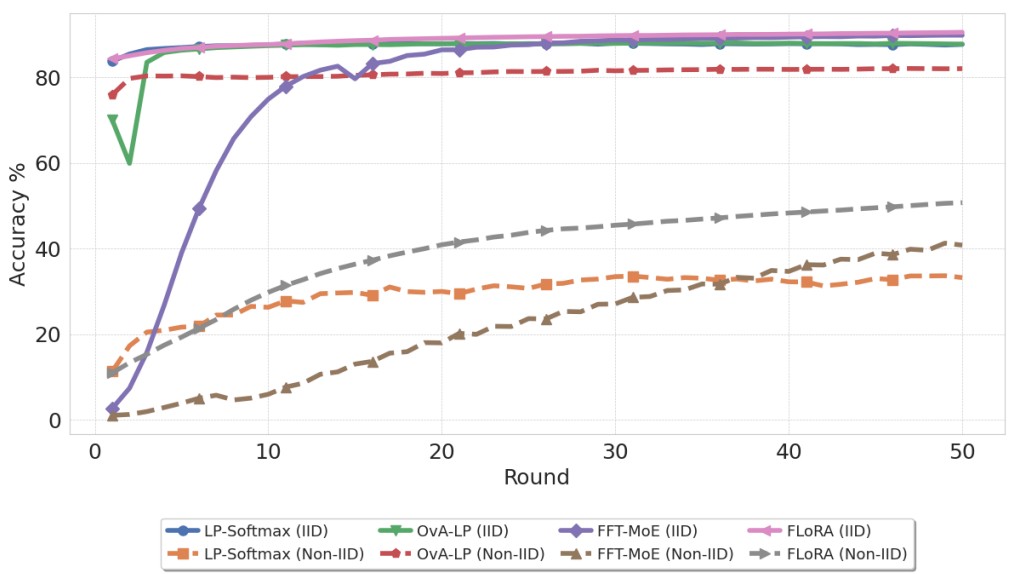

Figure 10: Baseline accuracy comparison under IID and Non-IID settings with participation ratio $p = 1.0$. Accuracy trajectories over 50 rounds.

| Methodology | Accuracy (%) | Acc@95 (Rounds) | Total Time (s) | Total Comm. (GB) |
|---|---|---|---|---|
| FFT-MoE (IID) | $89.81 \pm 0.00$ | $19 \pm 0$ | $11.70 \pm 0.00$ | $6.31 \pm 0.00$ |
| FFT-MoE (Non-IID) | $41.27 \pm 0.00$ | $47 \pm 0$ | $30.44 \pm 0.00$ | $15.61 \pm 0.00$ |
| FLoRA (IID) | $90.49 \pm 0.00$ | $4 \pm 0$ | $5.03 \pm 0.00$ | $131.43 \pm 0.00$ |
| FLoRA (Non-IID) | $50.73 \pm 0.00$ | $40 \pm 0$ | $47.57 \pm 0.00$ | $1314.27 \pm 0.00$ |
| LP-Softmax (IID) | $87.94 \pm 0.00$ | $1 \pm 0$ | $0.01 \pm 0.00$ | $0.06 \pm 0.00$ |
| LP-Softmax (Non-IID) | $33.67 \pm 0.00$ | $28 \pm 0$ | $0.38 \pm 0.00$ | $1.60 \pm 0.00$ |
| OvA-LP (IID) | $88.01 \pm 0.00$ | $4 \pm 0$ | $0.05 \pm 0.00$ | $0.23 \pm 0.00$ |
| OvA-LP (Non-IID) | $82.04 \pm 0.00$ | $2 \pm 0$ | $0.03 \pm 0.00$ | $0.11 \pm 0.00$ |

Table 5: Final performance of baselines on CIFAR-100 with participation ratio $p = 1.0$: accuracy, convergence rounds (Acc@95), and total costs until convergence.

| Methodology | Time (Client, ms) | Time (Server, ms) | Comm. (Client) | Comm. (Server) |
|---|---|---|---|---|
| FFT-MoE (IID) | $522.71 \pm 0.00$ | $93.33 \pm 0.00$ | 3.3999 MB | 339.99 MB |
| FFT-MoE (Non-IID) | $552.22 \pm 0.00$ | $95.38 \pm 0.00$ | 3.3999 MB | 339.99 MB |
| FLoRA (IID) | $348.50 \pm 0.00$ | $909.56 \pm 0.00$ | 336.4519 MB | 33645.19 MB |
| FLoRA (Non-IID) | $350.32 \pm 0.00$ | $839.02 \pm 0.00$ | 336.4519 MB | 33645.19 MB |
| LP-Softmax (IID) | $11.70 \pm 0.00$ | $1.62 \pm 0.00$ | 0.5867 MB | 58.67 MB |
| LP-Softmax (Non-IID) | $12.02 \pm 0.00$ | $1.63 \pm 0.00$ | 0.5867 MB | 58.67 MB |
| OvA-LP (IID) | $11.05 \pm 0.00$ | $1.62 \pm 0.00$ | 0.5867 MB | 58.67 MB |
| OvA-LP (Non-IID) | $11.63 \pm 0.00$ | $1.63 \pm 0.00$ | 0.5867 MB | 58.67 MB |

Table 6: Per-round computation and communication costs of baselines on CIFAR-100 with participation ratio $p = 1.0$.

Fig. 10 and Tables 5, 6 report baseline comparisons under the same configuration used in the ablation: CIFAR-100 with 100% participation, ViT-B/32 encoder, batch size 64, and measurement protocol defined in Appendix A.1. This setting represents an ideal regime in which encoder adaptation is not required.

Under IID, all methods except FFT-MoE converge to similar accuracy. Under Non-IID, however, OvA-LP reaches Acc@95 in only 2 rounds, compared to 28 rounds (LP-Softmax), 40 (FLoRA), and 47 (FFT-MoE), confirming that our framework is neither hyperparameter-dependent nor encoder-dependent.

OvA-LP also achieves near-IID performance under Non-IID while requiring drastically lower total cost: time is reduced by approximately $13\times$–$1600\times$, and communication by roughly $14\times$–$12000\times$ compared to existing baselines.

This best-case result illustrates how combining LP-level cost with significantly faster convergence yields extremely low end-to-end overhead in ideal settings, highlighting the upper bound of the practical efficiency that OvA-LP can provide.

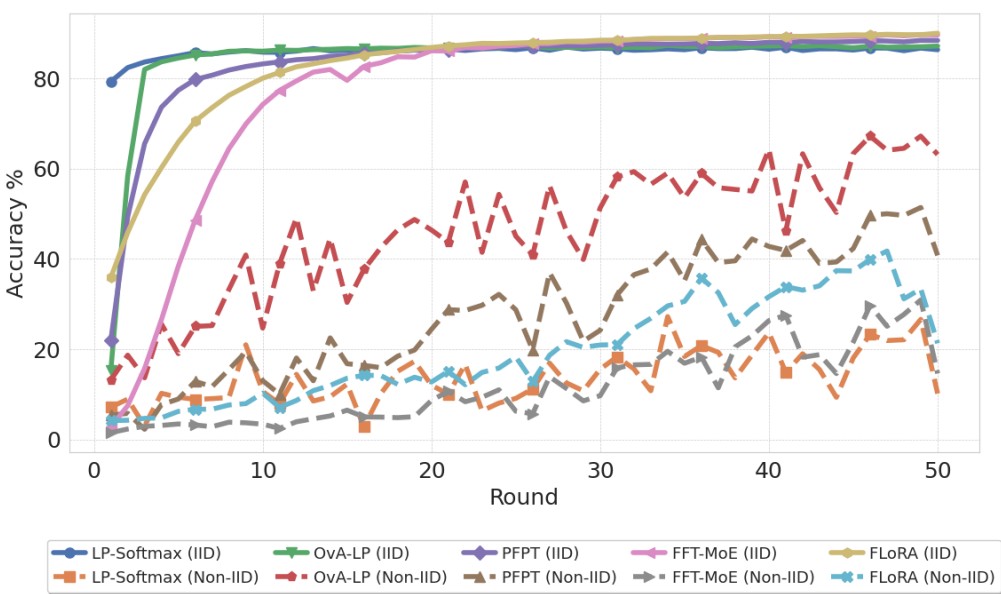

Figure 11: Baseline accuracy comparison under IID and Non-IID settings with participation ratio $p = 0.1$. Accuracy trajectories over 50 rounds.

| Methodology | Accuracy (%) | Acc@95 (Rounds) | Total Time (s) | Total Comm. (GB) |
|---|---|---|---|---|
| FFT-MoE $p = 0.1$ (IID) | $89.71 \pm 0.00$ | $20 \pm 0$ | $10.37 \pm 0.00$ | $0.66 \pm 0.00$ |
| FFT-MoE $p = 0.1$ (Non-IID) | $30.90 \pm 0.00$ | $46 \pm 0$ | $23.87 \pm 0.00$ | $1.53 \pm 0.00$ |
| PFPT $p = 0.1$ (IID) | $88.45 \pm 0.00$ | $12 \pm 0$ | $3789.42 \pm 0.00$ | $0.40 \pm 0.00$ |
| PFPT $p = 0.1$ (Non-IID) | $51.40 \pm 0.00$ | $46 \pm 0$ | $14532.63 \pm 0.00$ | $1.55 \pm 0.00$ |
| FLoRA $p = 0.1$ (IID) | $90.02 \pm 0.00$ | $17 \pm 0$ | $8.05 \pm 0.00$ | $55.86 \pm 0.00$ |
| FLoRA $p = 0.1$ (Non-IID) | $41.75 \pm 0.00$ | $46 \pm 0$ | $20.66 \pm 0.00$ | $151.14 \pm 0.00$ |
| LP-Softmax $p = 0.1$ (IID) | $86.96 \pm 0.00$ | $3 \pm 0$ | $0.04 \pm 0.00$ | $0.02 \pm 0.00$ |
| LP-Softmax $p = 0.1$ (Non-IID) | $27.20 \pm 0.00$ | $34 \pm 0$ | $0.41 \pm 0.00$ | $0.19 \pm 0.00$ |
| OvA-LP $p = 0.1$ (IID) | $87.39 \pm 0.00$ | $4 \pm 0$ | $0.05 \pm 0.00$ | $0.02 \pm 0.00$ |
| OvA-LP $p = 0.1$ (Non-IID) | $67.23 \pm 0.00$ | $40 \pm 0$ | $0.46 \pm 0.00$ | $0.23 \pm 0.00$ |

Table 7: Final performance of baselines on CIFAR-100 with participation ratio $p = 0.1$: accuracy, convergence rounds (Acc@95), and total costs until convergence.

Fig. 11 and Tables 7, 8 summarize baseline performance under a practical participation ratio of $p = 0.1$. As expected, reducing participation increases the number of rounds required for conver-

| Methodology | Time (Client, ms) | Time (Server, ms) | Comm. (Client) | Comm. (Server) |
|---|---|---|---|---|
| FFT-MoE $p = 0.1$ (IID) | $506.29 \pm 0.00$ | $12.08 \pm 0.00$ | 3.3999 MB | 34.00 MB |
| FFT-MoE $p = 0.1$ (Non-IID) | $506.81 \pm 0.00$ | $12.17 \pm 0.00$ | 3.3999 MB | 34.00 MB |
| PFPT $p = 0.1$ (IID) | $1734.80 \pm 0.00$ | $314049.79 \pm 0.00$ | 3.4539 MB | 34.54 MB |
| PFPT $p = 0.1$ (Non-IID) | $1648.06 \pm 0.00$ | $314278.61 \pm 0.00$ | 3.4539 MB | 34.54 MB |
| FLoRA $p = 0.1$ (IID) | $346.88 \pm 0.00$ | $126.40 \pm 0.00$ | 336.4519 MB | 3364.52 MB |
| FLoRA $p = 0.1$ (Non-IID) | $335.08 \pm 0.00$ | $113.97 \pm 0.00$ | 336.4519 MB | 3364.52 MB |
| LP-Softmax $p = 0.1$ (IID) | $11.96 \pm 0.00$ | $0.32 \pm 0.00$ | 0.5867 MB | 5.87 MB |
| LP-Softmax $p = 0.1$ (Non-IID) | $11.72 \pm 0.00$ | $0.32 \pm 0.00$ | 0.5867 MB | 5.87 MB |
| OvA-LP $p = 0.1$ (IID) | $11.09 \pm 0.00$ | $0.38 \pm 0.00$ | 0.5867 MB | 5.87 MB |
| OvA-LP $p = 0.1$ (Non-IID) | $11.19 \pm 0.00$ | $0.32 \pm 0.00$ | 0.5867 MB | 5.87 MB |

Table 8: Per-round computation and communication costs of baselines on CIFAR-100 with participation ratio $p = 0.1$.

gence across all methods: OvA-LP reaches Acc@95 in 40 rounds, compared to 34 for LP-Softmax, 46 for FLoRA, 46 for FFT-MoE, and 46 for PFPT. However, the final Non-IID accuracy differs substantially. OvA-LP achieves 67.23% under Non-IID, substantially outperforming LP-Softmax (27.20%) and all adaptation-based FFT methods (FLoRA, FFT-MoE, PFPT) in this setting.

In terms of total cost, OvA-LP converges in only $0.46\,\mathrm{s}$ and $0.23\,\mathrm{MB}$, while adaptation-based approaches require between $45\times$ and $4500\times$ more time and $7\times$ to $650\times$ more communication. LP-Softmax is cheaper in cost but severely underperforms in accuracy, reinforcing that OvA-LP provides a significantly better trade-off between robustness and efficiency when participation is limited.

Overall, under participation-restricted settings where encoder adaptation is unnecessary, OvA-LP offers a favorable accuracy–efficiency trade-off, maintaining substantially higher Non-IID accuracy than competing methods.

# B    BASELINE SETTINGS

## B.1    DATASET SETTINGS

| Work | Dataset | # Clients |
|---|---|---|
| FedProx (Li et al., 2020) | MNIST, FEMNIST, Sent140, Shakespeare | 10 |
| SCAFFOLD (Karimireddy et al., 2020) | EMNIST | 20 |
| FedLTF (Zhan et al., 2025) | CIFAR-10/100, MNIST/FMNIST | 20 |
| PFPT (Weng et al., 2024) | CIFAR-10/100, TinyImageNet | 10 |
| FFT-MoE (Hu et al., 2025) | CIFAR-10, AgNews | 4, 10 |
| **Our setup** | CIFAR-100, TinyImageNet, AgNews, EMNIST, DomainNet | **100** |

Table 9: Survey of FL benchmark settings in recent literature. Our setup scales to 100 clients and five datasets.

As shown in Table 9, prior studies typically evaluate on smaller client and dataset scales, whereas our configuration substantially increases both dimensions.

## B.2    BASELINE PARAMETERS

Table 10 summarizes the unified training configurations used across PFPT, FFT-MoE, and FLoRA. Model-specific hyperparameters follow their original implementations: PFPT uses 10 prompt tokens; FFT-MoE employs 8 experts with rank 2 per expert, top-1 routing, and auxiliary load-balancing loss coefficient $\lambda = 10^{-5}$; and FLoRA adopts LoRA-based adaptation with local epochs = 1, learning rate $3 \times 10^{-4}$, rank = 8, scaling factor $\alpha = 16$, dropout = 0.05, applied to QKV projection layers, using base rank schedule [64, 32, 16, 16, 8, 8, 4, 4, 4, 4] scaled by $\times 10$ for 100 clients.

| Method | Optimizer | Learning rate | Local epochs |
|---|---|---|---|
| LP-Softmax | AdamW (weight decay = $10^{-4}$) | $10^{-2}$ | 1 |
| OvA-LP | AdamW (weight decay = $10^{-4}$) | $10^{-2}$ | 1 |
| PFPT | Adam (weight decay = 0) | $10^{-4}$ | 5 |
| FFT-MoE | Adam (weight decay = $10^{-2}$) | $3 \times 10^{-4}$ | 1 |
| FLoRA | AdamW (weight decay = $10^{-2}$) | $3 \times 10^{-4}$ | 1 |

Table 10: Training configurations used for LP-Softmax, OvA-LP, PFPT, FFT-MoE, and FLoRA.

## USE OF LARGE LANGUAGE MODELS

We used large language models (e.g., ChatGPT) in two limited ways: (i) to polish the writing and improve readability, and (ii) to aid in the discovery of related work. No parts of the conceptual design, experiments, or analysis were generated by LLMs.

