# OpenReview forum: "OVA-LP: A Simple and Efficient Framework for Federated Learning on Non-IID Data"
_ICLR.cc/2026/Conference — Submitted to ICLR 2026_

### Official Review · Reviewer_4AKB · 2025-10-27

**Soundness:** 2
**Presentation:** 2
**Contribution:** 2
**Rating:** 4
**Confidence:** 3

**Summary:**

This paper introduces OvA-LP, a framework that combines linear probing, one-vs-all (OvA) heads, and a two-stage training schedule to mitigate client drift in Federated Fine-Tuning (FFT) of foundation models on non-IID data.

**Strengths:**

- **Strong Empirical Performance**.
- **High Efficiency**.
- **Simplicity and Modularity**.

**Weaknesses:**

1. **Incremental Contribution:** The core components (linear probing, OvA heads, two-stage training) are all established techniques. The paper fails to demonstrate novel synergistic mechanisms beyond their combination.
2. **Superficial Theoretical Analysis:** The bias-variance decomposition remains qualitative. Lacks formal proofs or bounds to substantiate claims about bias reduction or variance control.
3. **Limited Experimental Validation:**
    - Scope is narrow (only vision tasks, ViT encoders). No results on NLP/time-series or with CNN architectures.
    - Baseline comparisons are incomplete and potentially unfair (e.g., unequal participation rates vs. PFPT).
    - Missing sensitivity analysis (hyperparameters, non-IID severity).
4. **Poor Practicality:**
    - Unrealistic 100% client participation assumption. No solutions offered for partial participation.
    - Incomplete cost analysis (no communication costs across encoder sizes or edge device deployment).

**Questions:**

See above.

---

> ### Author Response · Authors · 2025-11-21
> **Weakness 1 — Incremental Contribution**
>
> # **Weakness 1 — Incremental Contribution**
>
> We agree that the individual components of OvA-LP — **linear probing, one-vs-all heads, and a two-stage schedule** — are established techniques. Our goal is not to propose a new primitive layer or optimizer, but to show that **a deliberately minimal and modular design is already sufficient to resolve severe Non-IID issues in FFT**, while remaining compatible with existing aggregation and personalization mechanisms.
>
> Most prior FFT methods we compare against (e.g., PFPT, FFT-MoE, FLoRA) address Non-IID challenges through **complex, tightly coupled pipelines**: encoder-side prompts or adapters, specialized aggregation rules, and personalized heads are co-designed and tuned together. In contrast, OvA-LP intentionally keeps the backbone and the FL protocol as simple and generic as possible:
>
> * we **freeze the shared encoder** and reuse standard FL aggregation (e.g., FedAvg),
> * we only modify the **classification head and training schedule**, and
> * these modifications are **orthogonal** to aggregation and personalization modules, so OvA-LP can in principle be combined with them.
>
> The key design idea is to view client drift in FFT as arising from **separable sources** and to address each source with a minimal head-only mechanism:
>
> * freezing the encoder and using LP preserves the **feature geometry** and removes representation drift in the backbone,
> * OvA heads **decouple class logits**, preventing label-skew-induced coupling in the head, and
> * the two-stage schedule (positive-only, then positive+negative) stabilizes **training dynamics** in the early rounds and enables fast convergence under heterogeneous client updates.
>
> Our existing ablation (Figure 3 in the original submission) is structured around this design. It compares:
>
> * **LP-Softmax** (LP + standard head, no OvA, no two-stage),
> * **OvA-LP w/o two-stage** (LP + OvA, single-stage), and
> * **full OvA-LP** (LP + OvA + two-stage).
>
> The results show that LP-Softmax is clearly the weakest in Non-IID, adding OvA **raises the attainable Non-IID accuracy ceiling** by removing label-skew effects, and adding the two-stage schedule on top **significantly accelerates and stabilizes convergence**, yielding the strongest Non-IID behavior among LP-based variants.
>
> We therefore do not claim conceptual novelty for LP, OvA, or two-stage training in isolation. The contribution of this work lies in:
>
> 1. **A minimal, head-only FFT design (OvA-LP) that is easy to implement and modular with respect to aggregation and personalization**, and
> 2. **Systematic empirical evidence** that this simple recipe can match or surpass more complex PEFT-based FFT methods on challenging Non-IID benchmarks.
>
> We have adjusted the way we describe OvA-LP in the paper to emphasize it as a **simple LP-style FFT method with strong empirical performance and broad modularity**, rather than as a completely new architectural primitive.

---

> > ### Author Response · Authors · 2025-11-21
> > **Weakness 3 — Limited Experimental Validation (1)**
> >
> > # **Weakness 3 — Limited Experimental Validation (scope, baselines, sensitivity)**
> >
> > **(a) Scope: “only vision tasks, ViT encoders”**
> >
> > We agree that the *original* submission focused primarily on vision tasks with pretrained encoders. In the revised version, we explicitly extended the experimental scope in two directions:
> >
> > 1. **A CNN backbone within vision**, and
> > 2. **A text classification task** with pretrained language encoders under the same FFT protocol.
> >
> > Section 4 now includes the following additional experiments for **OvA-LP** (same FFT setup: 100 clients, Non-IID partition, 50 rounds, head-only training):
> >
> > |    Backbone   | Dataset   | Setting                                | Relative Performance ($R(50)$, %) |
> > | :-----------: | :-------- | :------------------------------------- | :-----------------------------: |
> > | ConvNeXt-Base | CIFAR-100 | Non-IID FFT under shared configuration |              93.4%              |
> > |  RoBERTa-Base | AGNews    | Non-IID text classification (FFT)      |              84.2%              |
> > |  SBERT-MiniLM | AGNews    | Non-IID text classification (FFT)      |              91.0%              |
> >
> >
> > Here,
> > $R(50) = \text{Acc} _{\text{NonIID(50)}} / \text{Acc} _{ \text{IID(50)}} \times 100$
> >
> > These results show that OvA-LP:
> >
> > * remains highly robust with **a CNN backbone** (ConvNeXt-Base on CIFAR-100), and
> > * maintains strong relative performance in **Non-IID text classification** with both RoBERTa-Base and SBERT-MiniLM encoders,
> >
> > without changing the overall FFT recipe. This indicates that OvA-LP is **not a ViT-specific trick**, but a generally applicable head-only FFT method across multiple encoders and tasks.
> >
> > We do not claim to cover all modalities (e.g., time-series) in this work, but we believe these additions substantially broaden the empirical scope beyond “only vision + ViT.”
> >
> > ---
> >
> > **(b) Baseline comparisons and participation rates (PFPT)**
> >
> > To address concerns about incomplete or unfair baseline comparisons, we **re-ran all main baselines under a unified configuration** on CIFAR-100 Non-IID:
> >
> > * shared encoder: ViT-B/32 (frozen),
> > * 100 clients, 50 communication rounds, batch size 64,
> > * Non-IID partition: Dirichlet with (p = 0.1, $\alpha = 0.001$),
> > * participation rates: **0.1 and 1.0**.
> >
> > Under this setting, the **Non-IID relative performance** ($R(50)$) is:
> >
> > | Participation rate | LP-Softmax | OvA-LP | FLoRA | PFPT | FFT-MoE |
> > | ------------------ | ---------: | -----: | ----: | ---: | ------: |
> > | 1.0                |       38.0 |   93.4 |  55.6 |    – |    44.6 |
> > | 0.1                |       24.1 |   74.9 |  37.3 | 54.7 |    28.6 |
> >
> > All methods share **exactly the same encoder and FL configuration**; only the head/adapter design differs. In both participation settings, **OvA-LP achieves the highest Non-IID relative performance**, while LP-Softmax does *not* outperform PEFT baselines. This directly addresses the fairness issue in the main comparison.
> >
> > For **PFPT with participation 1.0**, we omit the run because of its impractical runtime: the per-round wall-clock time increases from about **34 seconds** (participation 0.1) to about **3,754 seconds** (participation 1.0), which would require roughly **100 hours per seed** for a 50-round run. We therefore report PFPT at participation 0.1, where it can be trained to convergence, and compare it fairly against OvA-LP and other baselines in that realistic regime.
> >
> > (continued)

---

> > ### Author Response · Authors · 2025-11-21
> > **Weakness 3 — Limited Experimental Validation (2)**
> >
> > **(c) Sensitivity analysis (hyperparameters, Non-IID severity)**
> >
> > We did not conduct a separate, exhaustive hyperparameter or Non-IID-severity sweep. Instead, our focus in this work is on whether **a simple, fixed recipe** can be robust across strong Non-IID settings.
> >
> > On the **hyperparameter side**, OvA-LP is intentionally lightweight:
> >
> > * it trains only a **single linear head** (with an OvA decomposition),
> > * we fix the optimizer to **AdamW**, the learning rate to **$10^{-2}$**, the weight decay to **$10^{-4}$**, and the local epochs to **1 per round** for OvA-LP across all datasets and encoders,
> > * and we do **not retune** these hyperparameters per dataset or per backbone.
> >
> > Despite this lack of tuning, OvA-LP shows consistently strong Non-IID performance **across diverse encoders (ViT, ConvNeXt, RoBERTa, SBERT) and tasks (vision and text)**, suggesting that its behavior is robust to hyperparameter choices. This is one of the motivations for its simple, head-only design.
> >
> > On the **Non-IID severity** side, rather than sweeping over many mild configurations, we chose to evaluate OvA-LP and all baselines under a set of **extremely strong stress-test partitions**, including:
> >
> > * **Shard-1 / Shard-2** label partitions,
> > * **clustering-based feature skew**,
> > * **Dirichlet with (p = 0.1, $\alpha$ = 0.001)** (FedCorr-style Bernoulli-thinned Dirichlet), and
> > * **Zipf(2.0)** label skew.
> >
> > To the best of our knowledge, this combination corresponds to **the most severe Non-IID regimes used in the FFT literature**. Our intention was to evaluate OvA-LP and baselines under **worst-case Non-IID stress**, rather than to finely sweep from mild to extreme severities.
> >
> > In this paper, we chose to:
> >
> > * keep OvA-LP’s hyperparameters **fixed and simple**, and
> > * test all methods across **multiple datasets and highly heterogeneous partitions**, including some of the most severe Non-IID configurations used in FFT.
> >
> > We believe this design provides a practically meaningful picture of OvA-LP’s robustness and generality, even without a full sensitivity sweep.

---

> ### Author Response · Authors · 2025-11-21
> **Weakness 2 — Superficial Theoretical Analysis**
>
> # **Weakness 2 — Superficial Theoretical Analysis / Bias–Variance Decomposition**
>
> We appreciate the concern about theoretical depth. It is correct that our paper does **not** provide formal proofs or generalization/optimization bounds for FFT with large encoders under Non-IID data. The bias–variance discussion in the paper is intended as a **conceptual lens to guide the design and interpretation of OvA-LP**, rather than as a formal theorem or bound.
>
> Concretely, when we refer to “bias” and “variance” in the paper, we use these terms to qualitatively separate different **sources of degradation** in FFT:
>
> * “bias” caused by representation shifts when the shared encoder is updated under heterogeneous client data,
> * “bias” caused by label-skew-induced coupling between logits in a multi-class head, and
> * “variance” in training dynamics induced by heterogeneous local updates in Non-IID settings.
>
> OvA-LP is motivated as a head-only design that **separately targets** these effects:
>
> * freezing the encoder and using LP avoids representation drift in the backbone,
> * OvA heads reduce label-skew effects by decoupling per-class logits, and
> * the two-stage schedule (positive-only, then positive+negative) is designed to stabilize early training before exposing the model to full positive/negative interactions.
>
> We agree that this reasoning is **qualitative**. Our main way of substantiating these claims is **empirical**, not theoretical. In particular, the ablation study on CIFAR-100 under the standard Non-IID setting (our original ViT-L/16, full-participation configuration) compares:
>
> * **LP-Softmax** (LP + standard head, no OvA, no two-stage),
> * **OvA-LP w/o two-stage** (LP + OvA, single-stage), and
> * **full OvA-LP** (LP + OvA + two-stage).
>
> This ablation shows that:
>
> * LP-Softmax is clearly the weakest in Non-IID;
> * adding OvA (OvA-LP w/o two-stage) **raises the attainable Non-IID accuracy ceiling** by removing label-skew effects;
> * adding the two-stage schedule on top (full OvA-LP) further **improves Non-IID performance and stabilizes convergence**, yielding the strongest behavior among LP-based variants.
>
> This ablation is our primary “evidence” for the bias–variance-oriented design: it demonstrates, in a controlled setting, how each component contributes to robustness under Non-IID.
>
> Beyond this, the paper includes **numerous additional experiments** (comparisons with PEFT baselines in unified settings, participation-rate sweeps, and domain-shift evaluations on CIFAR-100, DomainNet-Clipart/Painting, and EMNIST). These experiments are not intended as theoretical proofs, but as **complementary empirical analyses** to assess robustness and generality of OvA-LP across different datasets, participation regimes, and levels of domain shift.
>
> In summary, we do not position this work as providing a complete theoretical characterization of Non-IID FFT with large encoders. The bias–variance view in the paper is used as a **qualitative framework** to structure the design of OvA-LP and to motivate which components we include. The support we provide is **empirical rather than formal**, centered on ablation studies and extensive experiments, and we see a more rigorous theoretical treatment as an important direction for future work.

---

> ### Author Response · Authors · 2025-11-21
> **Weakness 4 — Practicality**
>
> ### **Weakness 4 — Practicality (participation, communication cost, deployment)**
>
> We appreciate the reviewer’s concerns regarding the practicality of our setting, especially the use of 100% client participation and the lack of explicit discussion of encoder size and edge deployment. We address these points below.
>
> ---
>
> **(a) Participation assumptions and partial participation**
>
> The original submission emphasized the 100% participation setting. In the revised version, we now make explicit that our conclusions do **not** rely solely on this regime.
>
> First, we **re-ran all main baselines** (LP-Softmax, OvA-LP, FLoRA, PFPT, FFT-MoE) on CIFAR-100 under a **unified configuration** with participation rates **0.1 and 1.0**, and moved these results into Section 4. All methods share the same encoder (ViT-B/32), FL configuration (100 clients, 50 rounds, batch size 64), and Non-IID partition (Dirichlet with (p = 0.1, $\alpha$ = 0.001)).
>
> Under this setting, the Non-IID relative performance ($R(50)$) is:
>
> | Participation rate | LP-Softmax | OvA-LP | FLoRA | PFPT | FFT-MoE |
> | ------------------ | ---------: | -----: | ----: | ---: | ------: |
> | 1.0                |       38.0 |   93.4 |  55.6 |    – |    44.6 |
> | 0.1                |       24.1 |   74.9 |  37.3 | 54.7 |    28.6 |
>
> OvA-LP achieves the highest Non-IID relative performance at **both** participation rates, while LP-Softmax does not dominate PEFT baselines. This shows that OvA-LP’s robustness is not restricted to the full-participation setting.
>
> Second, we added a **participation-rate sweep** for OvA-LP (0.1–1.0) and report the CIFAR-100 Non-IID results:
>
> | Participation rate | 0.1  | 0.2  | 0.3  | 0.4  | 0.5  | 0.6  | 0.7  | 0.8  | 0.9  | 1.0  |
> | ------------------ | ---- | ---- | ---- | ---- | ---- | ---- | ---- | ---- | ---- | ---- |
> | ($R(50)$) (%)        | 79.3 | 90.6 | 92.7 | 93.6 | 94.5 | 94.6 | 95.0 | 95.2 | 95.6 | 95.8 |
>
> These results show a **graceful degradation**: as the participation rate decreases, ($R(50)$) drops smoothly rather than collapsing. Even at participation rate 0.1, OvA-LP retains 79.3% of its IID accuracy, and from 0.2 onward it already exceeds 90%. For participation rates 0.7 and above, ($R(50)$) essentially plateaus near 95%. In other words, 100% participation is best seen as an **upper bound** on performance and convergence speed, but OvA-LP continues to outperform or match baselines even at low participation rates.
>
> ---
>
> **(b) Communication cost and encoder size**
>
> OvA-LP is designed so that **all communication and training in the FL loop happens at the head level**:
>
> 1. Each client first performs a **one-time forward pass of the encoder** on its local dataset to precompute and cache encoder features.
> 2. During federated training, clients **only update and communicate the OvA linear head**, not the encoder.
>
> In our FFT setup, the per-round communication payload is determined by the head size
>
> $$ \text{head params} = \text{encoder output dim} \times \text{num classes} ,$$
>
> rather than by the total number of encoder parameters. In practice, commonly used encoders such as ViT-B/32, ConvNeXt-Base, RoBERTa-Base, and SBERT-MiniLM all have **similar feature dimensions** (hundreds to low thousands), so the head size—and hence the communication cost—stays in a narrow range across these backbones. Changing to a larger encoder mostly affects the **one-time precomputation step**, not the per-round FL communication.
>
> We also provide a **detailed breakdown of communication and timing costs in the Appendix** for all methods (LP-Softmax, OvA-LP, FLoRA, PFPT, FFT-MoE), including:
>
> * client-side vs. server-side costs, and
> * per-round vs. total costs over all rounds.
>
> These tables make explicit how OvA-LP’s head-only communication compares to baselines that train larger prompt/adapter/MoE modules.
>
> ---
>
> **(c) Edge-device deployment**
>
> From the perspective of edge deployment, OvA-LP keeps the **on-device training loop lightweight**:
>
> * Each device runs the pretrained encoder **once** to obtain cached features, and
> * all subsequent FL rounds involve backpropagation only through a **small linear OvA head**.
>
> This significantly reduces the memory and compute footprint during FL compared to methods that backpropagate through encoder-side prompts, adapters, or MoE layers in every round. Combined with the head-only communication described above, this makes OvA-LP particularly suitable for **resource-constrained edge devices**, while still allowing the use of strong pretrained encoders.

---

### Official Review · Reviewer_PhgA · 2025-10-27

**Soundness:** 2
**Presentation:** 3
**Contribution:** 2
**Rating:** 4
**Confidence:** 3

**Summary:**

The paper studies the problem of federated fine-tuning on non-IID client data, where models drift due to data heterogeneity. It introduces **OvA-LP**, a simple framework that prevents this drift **at its source** rather than correcting it afterward. OvA-LP freezes the pretrained encoder for stable features, replaces softmax with one-vs-all (OvA) heads to remove label bias, and uses a two-stage training process to reduce variance. Experiments show that OvA-LP achieves accuracy close to the IID setting while remaining efficient and resilient under various forms of label noise.

**Strengths:**

**1. Source-Level Philosophy:**
The paper introduces a “source-level” perspective on preventing client drift. Instead of adjusting aggregation or adding personalization after the fact, OvA-LP stops drift before it begins. This proactive approach is supported by a clear bias–variance framing that connects theory with observed results.

**2. Strong Theoretical Motivation:**
OvA-LP is grounded in a bias–variance decomposition of federated gradients, showing how each component addresses a specific cause of drift. The frozen encoder reduces feature-skew bias, the OvA head removes label-skew bias and variance, and the two-stage training schedule manages optimization variance. This structure gives a clear and convincing rationale for why the method works.

**3. Minimal Yet Effective Design:**
The framework stands out for its simplicity and practicality. By combining a frozen encoder, linear probing, and OvA heads, it achieves strong results without the complexity of mixture-of-experts or prompt-tuning models. Its modular design also allows easy integration with existing federated learning schemes.

**4. Ablation Studies:**
Ablation studies show that each component contributes meaningfully to performance gains. Moreover, the inclusion of label noise experiments demonstrates the robustness of OvA-LP under real-world conditions.

**Weaknesses:**

**1. Experimental Setup Clarification (Major):**
It is unclear whether the comparisons between OvA-LP and the baselines (FFT-MoE, PFPT) are conducted on the same dataset and under identical settings. Although the paper mentions that baseline methods are reproduced using their original architectures and training protocols, Table 7 shows different datasets and client counts for OvA-LP compared to the baselines. Moreover, Table 8 reports an active client ratio of 10% for PFPT, while OvA-LP uses 100% participation. This discrepancy raises concerns about the fairness of the comparison, especially given that OvA-LP’s reported advantage (95.9% vs. 10.1% and 34.5%) could partly stem from this difference. Clarification on whether all methods were evaluated under identical conditions would strengthen the paper’s claims.

**2. Limited Generality:**
The paper claims that OvA-LP’s minimalist design is modular and compatible with aggregation and personalization frameworks, suggesting potential use across diverse FFT pipelines. However, its comparisons are limited to personalization-based methods like FFT-MoE and PFPT. Extending the approach to PEFT settings, such as FLoRA, FedSA-LoRA, or FRLoRA, would strengthen its generality and applicability beyond vision tasks.

**3. Potential Overclaim in Efficiency:**
The claim that OvA-LP achieves substantially faster convergence could partially stem from using a frozen encoder (no backprop through large layers). A fair comparison should equalize encoder freezing across baselines or report separate encoder vs. head timings.

**4. Notational Inconsistency (Minor):**
In Sections 3.1 and 3.2, the notation for $(x, y)$ is inconsistent. In Section 3.1, $x$ and $y$ represent input and label pairs from the data distribution, while in Section 3.2, $y$ denotes a second input sample in the alignment term, making the notation unclear.

**Questions:**

Can you provide any analytical result or empirical variance plots that quantify the “variance suppression” beyond accuracy curves?

---

> ### Author Response · Authors · 2025-11-21
> **Weakness 1 — Experimental Setup and Fairness of Comparisons**
>
> # **Weakness 1 — Experimental Setup and Fairness of Comparisons**
>
> We thank the reviewer for raising this important concern. Our understanding is that:
> (i) in the original submission it was not clear whether OvA-LP and the baselines (FFT-MoE, PFPT, etc.) were evaluated under exactly the same architectural and federated configurations (encoder, client count, participation ratio, etc.);
> (ii) in particular, the original versions of Tables 7 and 8 could be read as if OvA-LP used full participation while some baselines (e.g., PFPT) used only 10% active clients or different encoder sizes; and therefore
> (iii) the large gaps in relative performance (for example, 95.9% vs. 34.5% and 10.1%) might be partly driven by these configuration differences rather than by the intrinsic properties of OvA-LP.
>
> We agree that the initial presentation could create this impression. In the original version, some experiments reused each baseline’s own implementation settings (for example, PFPT/FFT-MoE with their original encoders and participation ratios), while OvA-LP was reported with a different encoder and full participation. The text did not clearly separate results coming from fully matched configurations versus those following the original papers. We understand that this heterogeneous setup can reasonably raise fairness concerns.
>
> To address this, we **re-ran and consolidated the key comparisons under a fully unified configuration**, which is now explicitly described in the revised manuscript (Sec. 4.1, Tables 5–8). For CIFAR-100, **all FFT methods now share exactly the same federated setting**:
>
> * **Encoder:** ViT-B/32 (shared across OvA-LP, LP-Softmax, PFPT, FFT-MoE, and FLoRA)
> * **Dataset:** CIFAR-100
> * **Number of clients:** 100
> * **Communication rounds:** 50
> * **Non-IID construction:** Dirichlet partition with $p = 0.1$, $\alpha = 0.001$
> * **Participation ratios:** 0.1 and 1.0 (reported separately)
> * **Batch size:** 64 (fixed across methods)
>
> Within this unified setting:
>
> * **LP-Softmax** is a pure linear probing baseline: a frozen ViT-B/32 encoder with a standard softmax head (no encoder adaptation).
> * **OvA-LP** also keeps the encoder completely frozen, but replaces the head with OvA classifiers and uses the two-stage training schedule (no encoder adaptation).
> * **PFPT, FFT-MoE, and FLoRA** are PEFT-style methods that attach prompts, experts, or LoRA modules to the same shared encoder and adapt those modules to the Non-IID data.
>
> We quantify Non-IID robustness via
>
> $$R(t) = \text{Acc} _{\text{NonIID(t)}} / \text{Acc} _{ \text{IID(t)}} \times 100$$
>
> and summarize the endpoint by $R(50)$. Under the unified CIFAR-100 setting (seed = 0), the Non-IID $R(50)$ values are:
>
> | Participation rate | FFT-MoE | FLoRA |  OvA-LP  | LP-Softmax | PFPT |
> | ------------------ | :-----: | :---: | :------: | :--------: | :--: |
> | 1.0                |   44.6  |  55.6 | **93.4** |    38.0    |   –  |
> | 0.1                |   28.6  |  37.3 | **74.9** |    24.1    | 54.7 |
>
> PFPT with full participation (1.0) is omitted because its per-round runtime in this unified setting is about 3,754 seconds, which would require roughly 100 hours per seed for 50 rounds and is therefore impractical to run to completion.
>
> These unified results directly address the fairness concern: **OvA-LP is evaluated under exactly the same encoder, Non-IID construction, participation ratios, and federated configuration** as LP-Softmax, FLoRA, PFPT, and FFT-MoE. Under these matched conditions, **OvA-LP consistently achieves the highest Non-IID robustness**:
>
> * $R(50)$ = \mathbf{93.4}$ at participation 1.0 (vs. 55.6 for FLoRA, 44.6 for FFT-MoE, 38.0 for LP-Softmax)
> * $R(50)$ = \mathbf{74.9}$ at participation 0.1 (vs. 54.7 for PFPT, 37.3 for FLoRA, 28.6 for FFT-MoE, 24.1 for LP-Softmax)
>
> The revised Sec. 4.1–4.2 explicitly state that these numbers are obtained under this unified configuration. **In summary, while the original presentation could make the main comparison appear heterogeneous, the revised manuscript now re-runs all key baselines under a single shared setup, and the CIFAR-100 results above show that OvA-LP’s advantage persists even under strictly identical conditions.**

---

> ### Author Response · Authors · 2025-11-21
> **Weakness 2 — Limited Generality and Missing PEFT Baselines (1)**
>
> # **Weakness 2 — Limited Generality and Missing PEFT Baselines**
>
> We sincerely thank the reviewer for this insightful comment. Our understanding of your concern is that: (i) while the paper presents OvA-LP as a minimalist, modular framework that should be compatible with aggregation and personalization families, the original experiments primarily compared against personalization-oriented FFT-MoE and PFPT; (ii) there was no explicit comparison against LoRA-based FFT methods such as FLoRA, FedSA-LoRA, or FRLoRA, which weakens the claim that OvA-LP is competitive within modern PEFT-based FFT pipelines; and (iii) most of our evidence came from vision benchmarks, leaving it unclear whether the observed robustness extends beyond ViT-based image encoders to other architectures and NLP tasks.
>
> We appreciate this comment regarding generality beyond personalization-oriented FFT models. We would like to clarify that FFT-MoE and PFPT are indeed PEFT-based approaches that operate on a frozen encoder while introducing personalized modules, as explicitly clarified in the revised paper. However, we agree that the lack of a direct comparison with a LoRA-based FFT method and the focus on ViT-based vision experiments left an important gap in the original submission.
>
> To address this, the revised manuscript (1) adds a LoRA-based FFT baseline (FLoRA) under the same unified FFT configuration as in our response to Weakness 1, and (2) extends the empirical scope to include a CNN backbone and NLP encoders. We summarize these changes below.
>
> ---
>
> ### 1. Adding a LoRA-Based FFT Baseline
>
> In Section 4.3, we now compare OvA-LP against a broader set of state-of-the-art FFT baselines that explicitly includes a LoRA-based method:
>
> * PFPT — prompt-based PEFT
> * FFT-MoE — mixture-of-experts routing
> * FLoRA — LoRA-based adaptation
> * LP-Softmax — standard linear probing baseline
>
> All of these methods are evaluated under the **same unified CIFAR-100 FFT configuration** described in our response to **Weakness 1** (shared ViT-B/32 encoder, identical FL setup, participation rates 0.1 and 1.0). Under this unified setting, the CIFAR-100 Non-IID relative performance at round 50,
> $R(50) = \text{Acc} _{\text{NonIID(50)}} / \text{Acc} _{ \text{IID(50)}} \times 100$
> shows that OvA-LP achieves ($R(50)$ = 74.9%) at participation 0.1 and ($R(50)$ = 93.4%) at participation 1.0, while FLoRA, PFPT, FFT-MoE, and LP-Softmax all obtain lower ($R(50)$) values under the same encoder and federated configuration (see the table in our response to Weakness 1).
>
> These additions directly address the reviewer’s request for PEFT baselines: we now include **FLoRA** as a canonical LoRA-based FFT method alongside PFPT and FFT-MoE, ensuring that OvA-LP is compared against representative PEFT approaches from the prompt, MoE, and LoRA families **under identical FFT conditions**.
>
> ---
>
> ### 2. Cross-Architecture and Task Generalization, Including NLP Encoders
>
> To move beyond a single vision encoder, the revised Section 4.4.3 evaluates OvA-LP on **ConvNeXt-Base** (CIFAR-100) and **RoBERTa-Base / SBERT-MiniLM** (AGNews), explicitly moving into CNN and NLP settings. As summarized in the text, these experiments achieve relative performance in the range 84.2%–93.4%, showing that OvA-LP’s source-level design is not tied to ViT-based image encoders.
>
> For clarity, the backbones and tasks used in the revised experiments can be summarized as:
>
> |    Backbone   | Dataset   | Setting                                | Relative Performance ($R(50)$, %) |
> | :-----------: | :-------- | :------------------------------------- | :-------------------------------: |
> | ConvNeXt-Base | CIFAR-100 | Non-IID FFT under shared configuration |               93.4%               |
> |  RoBERTa-Base | AGNews    | Non-IID text classification (FFT)      |               84.2%               |
> |  SBERT-MiniLM | AGNews    | Non-IID text classification (FFT)      |               91.0%               |
>
> In all three cases, OvA-LP retains between 84.2% and 93.4% of the corresponding IID accuracy, confirming that the relative Non-IID robustness observed for ViT-based vision encoders extends to a CNN backbone and to NLP encoders as well, without changing the core OvA-LP recipe.
>
> (continued)

---

> ### Author Response · Authors · 2025-11-21
> **Weakness 2 — Limited Generality and Missing PEFT Baselines (2)**
>
> **In summary**, we agree that the original submission did not fully support the claims of generality and compatibility with PEFT-based FFT. In the revised manuscript, we:
>
> 1. **Expanded the FFT baselines to include FLoRA**, a LoRA-based PEFT method, and compared it against OvA-LP, PFPT, FFT-MoE, and LP-Softmax under the unified CIFAR-100 configuration presented in our response to Weakness 1; and
> 2. **Demonstrated cross-architecture and task robustness** by evaluating OvA-LP on ConvNeXt-Base and RoBERTa-Base / SBERT-MiniLM (AGNews), where all models retain 84.2%–93.4% of their IID accuracy.
>
> We hope these additions more concretely substantiate our generality claims: **OvA-LP’s source-level design remains competitive against representative PEFT-based FFT methods and exhibits consistent Non-IID robustness across both vision and NLP backbones**.

---

> ### Author Response · Authors · 2025-11-21
> **Weakness 3 — Potential Overclaim in Efficiency**
>
> # **Weakness 3 — Potential Overclaim in Efficiency**
>
> We appreciate the reviewer’s careful scrutiny of our efficiency claims. Our understanding of your concern is that: (i) OvA-LP’s substantially faster convergence might simply come from using a frozen encoder (i.e., avoiding backpropagation through large backbone layers), and (ii) a fair comparison should therefore either equalize encoder freezing across baselines or report separate encoder vs. head timings so that the source of the efficiency gain is transparent.
>
> We agree that the original wording did not make this separation as clear as it should have. In the revised manuscript we clarify both the **training regime** and the **scope** of our efficiency claim.
>
> ---
>
> ### 1. Encoder Freezing Is Already Equalized Across All Compared Methods
>
> First, we would like to explicitly clarify that **all methods in our experiments already share the same frozen-encoder regime**:
>
> * **PFPT, FFT-MoE, and FLoRA**: these methods attach prompts, experts, or LoRA adapters to a shared pretrained encoder and **only update those PEFT modules**. The encoder parameters themselves are kept fixed in our implementation.
> * **LP-Softmax and OvA-LP**: both keep the encoder frozen and **train only a linear head** (standard softmax vs. OvA with two-stage training), with **no additional PEFT modules** attached.
>
> In other words, there is **no asymmetric advantage** where only OvA-LP freezes the encoder while baselines fully fine-tune it; encoder freezing is already equalized across all methods in our unified experimental setup. The suggestion to “equalize encoder freezing across baselines” would, in our setting, mean freezing not only the encoder but also the PEFT modules of PFPT/FFT-MoE/FLoRA. This would effectively collapse them into a simple linear head on top of the frozen encoder, which is exactly what our **LP-Softmax** baseline represents and what we already compare against.
>
> ---
>
> ### 2. Design Trade-off: No PEFT Modules, Efficiency, and Explicitly Stated Limitations
>
> Second, we agree with the reviewer that **OvA-LP gains efficiency by using only a single linear OvA head, without any prompt/expert/LoRA modules, on top of the frozen encoder**. This naturally reduces computation and communication compared to methods that train additional PEFT modules.
>
> However, we view this not as an **unfair advantage**, but as a **deliberate design trade-off**:
>
> > OvA-LP is a fixed-encoder, no-PEFT FFT branch that achieves efficiency and stability by training only a linear OvA head, while intentionally giving up encoder/representation adaptiveness.
>
> In the revised manuscript, we make this trade-off and its consequences explicit in the **Discussion**, in line with the “scope of applicability” concern raised by Reviewer 2VP7. Concretely, we:
>
> * state that OvA-LP assumes a **strong pretrained encoder whose features are already suitable for (approximately) linear separation**, and
> * use the domain-shift experiments (CIFAR-100, DomainNet-Clipart/Painting, EMNIST) to spell out **where this design is appropriate and where it is not**:
>
>   * on **CIFAR-100** and **DomainNet-Clipart/Painting** (no or moderate domain shift), OvA-LP attains the strongest Non-IID robustness among all compared methods under the unified setting;
>   * on **EMNIST** (severe domain shift), **PEFT methods such as FLoRA outperform OvA-LP**, which we highlight as the downside of relying solely on a linear head without any PEFT modules.
>
> Thus, we do not present OvA-LP’s faster convergence as a universally fair “win” over all fine-tuning strategies. Rather, we explicitly frame it as:
>
> * **efficient and stable in fixed-encoder FFT regimes where a strong pretrained encoder is available**, and
> * **potentially suboptimal in regimes that require encoder or PEFT-module adaptation**, which is now clearly acknowledged in the Discussion.
>
> ---
>
> **In summary**, OvA-LP’s efficiency advantage in our experiments does **not** arise from being the only method that freezes the encoder, since **all compared methods already share the same frozen-encoder regime**. Instead, it comes from a **transparent design choice**: OvA-LP attaches **no PEFT modules** and trains only a linear OvA head, trading away adaptiveness for stability and efficiency. The revised manuscript makes this trade-off and its **limitations under severe domain shift (e.g., EMNIST, where PEFT methods become preferable)** explicit, and presents OvA-LP as an efficient fixed-encoder FFT option rather than as an unqualifiedly superior fine-tuning method in all settings.

---

> ### Author Response · Authors · 2025-11-21
> **Weakness 4 — Notation Consistency**
>
> #  **Weakness 4 — Notation Consistency**
>
> Thank you for pointing out the notation issue. In the revised Section 3.2, we now use **($x_i$,$x_j$)** for paired input samples and reserve **($x$, $y$)** exclusively for input–label pairs, eliminating the previous ambiguity.

---

> ### Author Response · Authors · 2025-11-21
> **Question — Variance Suppression**
>
> # **Question — Variance Suppression**
>
> Thank you for the insightful question. We apologize if our terminology caused confusion regarding the handling of variance. We would like to clarify that the paper does **not** claim to explicitly reduce variance in a quantitative/statistical sense, nor do we optimize any explicit variance-related objective.
>
> The phrase **“bias–variance suppression design”** may have suggested a stronger claim than intended. Our goal was to provide a **structural interpretation framework** for how OvA-LP mitigates the *causes* of destabilizing drift, rather than to claim direct numerical suppression of variance itself. Concretely:
>
> * **Feature-skew-induced bias** is alleviated by preserving the pretrained feature geometry through the LP structure.
> * **Label-skew-induced bias** is mitigated by OvA decoupling, which removes cross-class coupling in the head.
>
>   * Clarification: By removing these bias sources and couplings (e.g., cross-class covariance in a softmax head), we aim to prevent *amplification* of instability, but we do **not** claim to reduce or suppress the underlying variance itself.
> * The **two-stage schedule** is designed to stabilize optimization in the presence of residual variance.
>
>   * Clarification: This stage-wise curriculum does **not** attempt to reduce or suppress the remaining variance; instead, it helps the model **withstand** that variance and still converge stably under Non-IID updates.
>
> As we state in the paper, our view is that
>
> > variance cannot be eliminated entirely, but its destabilizing effect can be controlled through a two-stage curriculum aligned with the OvA structure.
>
> Accordingly, our empirical analysis focuses on **how stably and how fast models converge under Non-IID data, and what accuracy they finally attain**, rather than on plotting explicit variance statistics. We fully acknowledge that variance itself remains; OvA-LP’s contribution is to **remove bias-driven amplification mechanisms and to maintain stable convergence despite the residual variance**, rather than to quantitatively minimize or suppress variance in a strict statistical sense.

---

### Official Review · Reviewer_2VP7 · 2025-11-05

**Soundness:** 2
**Presentation:** 3
**Contribution:** 2
**Rating:** 6
**Confidence:** 4

**Summary:**

The paper introduces OvA-LP, a framework for federated fine-tuning (FFT) on non-IID data. It proposes to suppress local drift "at its source" by combining three components: (1) linear probing (LP) on a completely frozen encoder; (2) a one-vs-all (OvA) classification head to decouple logits ; and (3) a two-stage (positive-only, then positive+negative) training schedule. The proposed method is simple, efficient , and demonstrates exceptionally strong empirical results in its primary setting (e.g., 95.9% IID accuracy retention vs. 10.1%-34.5% for SOTA baselines). The clarity of the bias-variance motivation is also a strength.

**Strengths:**

1. **Clear Motivation:** The design is well-grounded in a bias-variance decomposition. Each component is clearly justified as targeting a specific source of drift: linear probing limits feature-skew bias , OvA heads eliminate label-skew bias , and the two-stage training controls variance.

2. **Methodological Simplicity**: The paper presents a "minimalist" framework that combines three relatively simple components: linear probing on a frozen encoder, one-vs-all (OvA) heads, and a two-stage training schedule.

3. **High Efficiency and Fast Convergence**: The framework is computationally efficient. By precomputing encoder features, the per-round training cost is nearly independent of the encoder's size. Furthermore, it converges significantly faster than baselines, often reaching near-IID performance in just 1-3 rounds.

4. **Very Nice Reproducibility:** The code implementation fully covers all the core methods, experimental settings, ablation studies, and baseline comparisons described in the paper. The codebase structure is clear and corresponds one-to-one with the arguments in the paper. This also helps in understanding the technical details in the paper. I personally recommend that other reviewers take this into consideration when evaluating the paper.

**Weaknesses:**

1. **Unfair Baseline Comparison**: This is the most critical issue. The paper compares OvA-LP- which only trains a linear head on a frozen encoder - against SOTA baselines like PFPT and FFT-MoE. These baselines are Parameter-Efficient Fine-Tuning (PEFT) methods that adapt the model (e.g., via prompt-tuning or MoE layers).
The "local drift" that the paper claims to solve is a phenomenon that arises precisely because the baselines are fine-tuning the shared encoder on heterogeneous local data. OvA-LP does not solve this drift; it avoids the problem entirely by not fine-tuning the encoder at all.
Therefore, the comparison in Figure 4 is an unfair comparison between a non-adapting model (OvA-LP) and adapting models (PFPT, FFT-MoE). The conclusion that OvA-LP is a superior framework for "federated fine-tuning" is unsupported by this evidence.
The paper's own ablation study (Figure 3) demonstrates this issue. The "LP-Softmax" baseline (frozen encoder + standard head) achieves 56.3% relative performance. This frozen-encoder baseline already massively outperforms the fine-tuning SOTA baselines (PFPT at 34.5%, FFT-MoE at 10.1%). The paper even states, "even LP-softmax... already surpasses post-hoc baselines".
This strongly implies that the primary reason for the dramatic performance gain (from 10-35% to 56%+) is freezing the encoder, not the specific OvA-LP design. While the OvA-LP components provide a further, significant boost (from 56.3% to 95.9%), the paper's primary comparison against fine-tuning methods is misleading. The authors must re-frame the contribution: OvA-LP is a superior linear probing method, not a superior fine-tuning method.

2. **Mismatch in Claims and Methodology**: The paper positions itself within the "PEFT-based FFT paradigm". Linear probing is the most trivial form of PEFT and is functionally distinct from methods like LoRA, prompt tuning (like PFPT), or adapters, which are designed to adapt the foundation model. Calling this "federated fine-tuning" is an overclaim. This is "federated linear probing." This distinction is critical because the method sacrifices model adaptation to gain robustness. This trade-off is not adequately discussed.

3. **Practicality of Experimental Assumptions**: The main results rely on 100% client participation , which is an unrealistic assumption for most practical FL scenarios. The paper acknowledges this in the limitations and Appendix A.3 , where it is shown that "Lower participation ratios lead to slower convergence". This weakens the main paper's claims of extreme efficiency and rapid convergence (e.g., "1 round" to Acc@95 in Table 3 ). Results under partial participation shall be moved to the main paper and analyzed thoroughly.

**Questions:**

1. **Revise Baseline Comparisons**: To make a fair comparison, the authors must compare OvA-LP to other frozen-encoder baselines. The current SOTA baselines (PFPT, FFT-MoE) must be re-run in a configuration where their encoders are also frozen, to isolate the benefit of their respective heads/adapters. Alternatively, the authors must compare OvA-LP against a true SOTA federated fine-tuning method (e.g., Fed-LoRA) and honestly discuss the trade-off (i.e., OvA-LP gains non-IID robustness by losing the ability to adapt the encoder).

2. **Discuss Scope of Applicability**: The method inherently relies on a high-quality pretrained encoder whose "feature geometry" is already sufficient for linear separation. This is a major assumption. The limitations section should be expanded to more critically discuss what happens when the downstream task is not linearly separable from the pretrained features and requires encoder adaptation (a scenario where OvA-LP would presumably fail completely).

---

> ### Author Response · Authors · 2025-11-21
> **Weakness 1 — Unfair Baseline Comparison (1)**
>
> # **Weakness 1 — Unfair Baseline Comparison**
>
> We appreciate the reviewer’s detailed discussion of this point. Our understanding of your concern is that: (i) OvA-LP only trains a linear head on a frozen encoder, while PFPT and FFT-MoE are PEFT-based methods that adapt the model (via prompts or MoE layers); (ii) in the original submission, LP-Softmax appeared to already outperform these “fine-tuning SOTA” baselines (56.3% vs 34.5% and 10.1% relative performance), which suggests that most of the gain comes from freezing the encoder rather than from the specific OvA-LP design; and therefore (iii) Figure 4 may be interpreted as an unfair comparison between a non-adapting model (OvA-LP) and adapting models (PFPT, FFT-MoE), making the claim that OvA-LP is a superior framework for federated fine-tuning appear unsupported.
>
> We agree that the **original configuration and phrasing made this interpretation possible**. In the initial submission, LP-Softmax used a ViT-L/16 encoder, while PFPT and FFT-MoE followed their original implementations with smaller encoders (e.g., ViT-B/32 or ViT-B/16). In addition, the sentence “even LP-softmax already surpasses post-hoc baselines” could be read as suggesting that a frozen LP baseline inherently dominates PEFT methods. In hindsight, this heterogeneous encoder choice and wording made it natural to attribute the dramatic performance gap primarily to encoder freezing rather than to the OvA head and training schedule.
>
> To address this, we first recall our evaluation philosophy for Non-IID robustness. For each round (t), we measure
>
> $$R(t) = \text{Acc} _{\text{NonIID(t)}} / \text{Acc} _{ \text{IID(t)}} \times 100$$
>
> and we summarize the endpoint gap by the final ratio ($R(50)$). This provides a unified lens for how closely each method tracks its IID trajectory in terms of both convergence and stability, rather than comparing absolute accuracies in isolation.
>
> Focusing on **CIFAR-100**, we reran all methods under a **fully unified configuration** to remove any encoder-related confounders:
>
> * Encoder: ViT-B/32 (shared across all methods)
> * Dataset: CIFAR-100
> * Number of clients: 100
> * Communication rounds: 50
> * Batch size: 64
> * Non-IID construction: Dirichlet partition (p = 0.1, α = 0.001)
> * Participation ratios: 0.1 and 1.0
>
> Under this setting:
>
> * **LP-Softmax** is a pure linear probing baseline: a frozen ViT-B/32 encoder with a standard softmax head (no encoder adaptation).
> * **OvA-LP** also keeps the encoder completely frozen, but replaces the head with OvA classifiers and trains them with the two-stage schedule (again, no encoder adaptation).
> * **PFPT / FFT-MoE / FLoRA** follow PEFT-style implementations that attach prompts, experts, or LoRA modules to the shared encoder and **adapt these modules** to the Non-IID data (adapting methods in the reviewer’s terminology).
>
> The table below reports the **CIFAR-100 Non-IID ($R(50)$)** values under this unified setup (seed = 0):
>
> | Part. Rate | FFT-MoE | FLoRA | OvA-LP | LP-Softmax | PFPT |
> | ---------: | :-----: | :---: | :----: | :--------: | :--: |
> |        1.0 |   44.6  |  55.6 |  93.4  |    38.0    |   –  |
> |        0.1 |   28.6  |  37.3 |  74.9  |    24.1    | 54.7 |
>
> PFPT with full participation (1.0) is omitted due to impractical runtime, which grows nonlinearly (34s → 3,754s per round; about 100 hours per seed).
>
> From this table, we observe:
>
> 1. **LP-Softmax no longer dominates the PEFT baselines.**
>    In contrast to the original heterogeneous-encoder setting, the unified ViT-B/32 results show that LP-Softmax is actually **weaker** than PEFT methods in terms of Non-IID robustness:
>
>    * At participation 1.0: LP-Softmax (38.0) < FFT-MoE (44.6) < FLoRA (55.6).
>    * At participation 0.1: LP-Softmax (24.1) is strictly below PFPT (54.7), FLoRA (37.3), and FFT-MoE (28.6).
>      Thus, once encoder size and configuration are controlled, a frozen LP baseline does **not** “massively outperform” PFPT and FFT-MoE. The impression that “freezing alone” explains the gap was largely an artifact of the heterogeneous encoder choices in the original submission.
>
> 2. **OvA-LP is the only method that remains extremely robust under Non-IID, despite using the same frozen encoder as LP-Softmax.**
>    OvA-LP shares the **exact same frozen ViT-B/32 encoder** as LP-Softmax, but with OvA heads and the two-stage training schedule. Under this unified setting, it achieves:
>
>    * ($R(50)$ = 93.4) at participation 1.0 (vs 55.6 for FLoRA, 44.6 for FFT-MoE, 38.0 for LP-Softmax),
>    * ($R(50)$ = 74.9) at participation 0.1 (vs 54.7 for PFPT, 37.3 for FLoRA, 28.6 for FFT-MoE, 24.1 for LP-Softmax).
>
>    Since OvA-LP and LP-Softmax both completely freeze the encoder, these results cannot be attributed to encoder freezing per se. Instead, they indicate that the **synergistic combination of the OvA head and the two-stage training schedule** is crucial for Non-IID robustness: simple LP with a frozen encoder is not enough.
>
> (continued)

---

> ### Author Response · Authors · 2025-11-21
> **Weakness 1 — Unfair Baseline Comparison (2)**
>
> These unified results also directly address the conceptual misunderstanding that motivated your concern: namely, that **local drift arises “precisely because” baselines fine-tune the shared encoder**, and that OvA-LP merely “avoids” drift by not fine-tuning. In our formulation, local drift stems from heterogeneous local updates, and these updates can occur in any trainable component—full encoder, PEFT modules, or even a linear head. The CIFAR-100 experiments above show that:
>
> * A **non-adapting** LP baseline (frozen encoder + standard head) still suffers from severe degradation in the Non-IID setting (low ($R(50)$) at both participation ratios).
> * Another **non-adapting** method, OvA-LP (frozen encoder + OvA heads + two-stage schedule), is instead highly robust.
>
> If local drift were purely a consequence of encoder adaptation, we would expect the non-adapting LP baseline to be inherently robust, which is contradicted by these results. Rather, the data suggest that **drift and instability persist even without encoder updates**, and that what matters is **how the trainable components (head or PEFT modules) interact with heterogeneous data**. OvA-LP does not “solve” drift by stepping outside the game; it remains in the same frozen-encoder regime as LP, but reduces drift through the synergistic design of its OvA heads and two-stage training.
>
> In summary, we agree that the original heterogeneous-encoder setup and wording could make it appear that the main performance gain came from encoder freezing, and that LP-Softmax alone already dominated PEFT baselines. To correct this, we reran CIFAR-100 under a unified ViT-B/32 configuration, added the explicit LP-Softmax baseline, and reported the relative robustness ($R(50)$). Under these controlled conditions, LP-Softmax does **not** outperform PFPT/FFT-MoE/FLoRA, while OvA-LP—using the **same frozen encoder** as LP—shows substantially higher Non-IID robustness. This directly addresses the concern that the comparison is unfair or that the observed gains are solely due to freezing, and clarifies that the core contribution of OvA-LP lies in the synergistic head design and training strategy rather than in avoiding encoder adaptation.

---

> ### Author Response · Authors · 2025-11-21
> **Weakness 2 — Mismatch in Claims and Methodology**
>
> # **Weakness 2 — Mismatch in Claims and Methodology (“PEFT-based FFT paradigm” vs federated linear probing)**
>
> We understand the reviewer’s concern as follows: although the paper positions OvA-LP within the “PEFT-based FFT paradigm”, the method itself belongs to the most trivial end of this spectrum (linear probing with a frozen encoder and learned head), and is functionally different from encoder-adapting PEFT methods such as LoRA, prompts, or adapters. Thus, calling OvA-LP a “federated fine-tuning framework” may appear over-claiming, and the trade-off of **sacrificing encoder adaptation to gain robustness** is not sufficiently highlighted.
>
> We agree that our original submission did not clearly disentangle the **different branches of FFT** and our own position within them. In the **revised manuscript**, we now explicitly organize the FFT landscape into three regimes:
>
> * **Full fine-tuning**, which updates all (or most) encoder parameters.
> * **PEFT**, which freezes the encoder but adapts lightweight modules (adapters, LoRA, prompts, experts, etc.).
> * **Linear probing (LP)**, which updates only a linear head on top of a frozen encoder and therefore *“lacks any adaptiveness”* and has traditionally been regarded as inadequate for demanding downstream tasks.
>
> OvA-LP is placed **explicitly in this LP branch**: it is a minimalist FFT framework that combines LP with OvA heads and a two-stage schedule, not a method that adapts encoder parameters or PEFT modules.
>
> In addition, recent work on parameter-efficient tuning in visual recognition, *“Lessons and Insights from a Unifying Study of PEFT in Visual Recognition”* (CVPR 2025), classifies **head-only tuning (including LP)** as a legitimate member of the fine-tuning family. Although that study is conducted in a standard (non-federated) setting, it provides independent evidence that LP is commonly treated as a **head-only fine-tuning method**: despite being the most minimal and “trivial” form, it still performs task-specific adaptation of classifier parameters on top of a frozen encoder. We leverage this perspective as external support for our terminology: in our work, “federated fine-tuning” is used in this broader sense, and OvA-LP is explicitly presented as an LP-style fine-tuning method deployed in a federated setting, distinct from encoder-adapting PEFT approaches.
>
> To avoid any impression of overclaiming, we have also clarified what OvA-LP **does and does not claim**:
>
> * We **do not** claim that OvA-LP universally dominates encoder-adapting PEFT methods (PFPT, FFT-MoE, FLoRA) across all tasks and domain shifts.
> * Instead, OvA-LP is positioned as a **minimalist, LP-based alternative within the FFT landscape**, designed for regimes where
>   (i) a strong pretrained encoder already provides reasonably structured representations, and
>   (ii) **Non-IID drift and variance**, rather than feature learning, are the dominant bottlenecks.
>
> In line with the reviewer’s *“Discuss Scope of Applicability”* comment, the **Discussion** section of the revised manuscript now state explicitly that OvA-LP **relies on a high-quality pretrained encoder whose feature geometry is already close to linearly separable**, and that this is a major assumption. When the downstream task is not linearly separable from the pretrained features and genuinely requires encoder adaptation, OvA-LP can indeed fail or be suboptimal compared to PEFT methods. We highlight this behavior empirically as well:
>
> * On standard or moderately shifted visual domains (e.g., CIFAR-100, DomainNet-Clipart/Painting), OvA-LP offers strong robustness and stability under Non-IID heterogeneity.
> * Under the most severe domain shift (EMNIST), **PEFT-based approaches remain beneficial**, and OvA-LP no longer outperforms them.
>
> These findings are described in the revised **Discussion and Conclusion**, which “suggest that the practical value of encoder adaptation depends on the severity of domain shift, and that relying solely on adaptation is not universally advantageous even in federated fine-tuning,” while simultaneously acknowledging that **OvA-LP itself is limited** when the encoder’s feature geometry is not suitable for linear probing.
>
> In summary, the revised manuscript:
>
> * clearly places OvA-LP in the **LP branch of the fine-tuning spectrum**,
> * uses the CVPR 2025 PEFT study as external evidence that LP is commonly regarded as a (head-only) fine-tuning method, and
> * makes the **trade-off explicit**: OvA-LP deliberately forgoes encoder adaptiveness in order to control drift and variance at the source, and is intended for regimes where this trade-off is favorable.
>
> A more detailed, experiment-driven discussion of this **scope of applicability and domain-shift behavior** (including EMNIST vs. DomainNet) is provided in our response to the reviewer’s “Discuss Scope of Applicability” question.

---

> ### Author Response · Authors · 2025-11-21
> **Weakness 3 — Practicality of Experimental Assumptions**
>
> # **Weakness 3 — Practicality of Experimental Assumptions (100% participation, “1 round” claims)**
>
> We thank the reviewer for pointing out this important issue. As you note, the main results in the original submission emphasized **100% client participation**, and Appendix A.3 only briefly showed that “lower participation ratios lead to slower convergence.” This naturally **weakens the main paper’s claims about extreme efficiency and rapid convergence** (e.g., “1 round” to Acc@95 in Table 3) in realistic FL scenarios where partial participation is the norm.
>
> We agree that full participation is not a realistic default assumption in mobile/edge FL. In the **revised manuscript**, we now treat **100% participation explicitly as an *upper-bound scenario***: the “1-round convergence” result is described as the **ideal upper bound under full participation**, and we no longer base our efficiency claims solely on this setting. Instead, all efficiency and robustness claims are now grounded in **partial participation experiments** as well.
>
> First, we **updated the main comparison in Section 4**. As detailed in our response to Weakness 1, we reran all baselines (LP-Softmax, OvA-LP, FLoRA, PFPT, FFT-MoE) under a unified configuration and evaluated them at **participation rate 0.1 and 1.0**. These new results **replace** the original Figure 4 and are now part of the main text. In both participation settings, OvA-LP remains the most robust method in terms of Non-IID $R(50)$, while LP-Softmax and PEFT baselines (FLoRA, PFPT, FFT-MoE) preserve the same relative ordering as in the full-participation case. Thus, the core robustness conclusion is now supported **simultaneously at participation rates 0.1 and 1.0**, rather than relying only on full participation.
>
> Second, we **recomputed cost and communication** for these more realistic settings. For CIFAR-100 with **participation rate 0.1** in the Non-IID setting, OvA-LP reaches Acc@95 in only **0.46 seconds** and **0.23 MB** of total communication. In contrast, adaptation-based FFT methods (FLoRA, FFT-MoE, PFPT) require between **45× and 4500× more time** and **7× to 650× more communication** to reach the same target accuracy. LP-Softmax is cheaper but severely underperforms in accuracy, reinforcing that OvA-LP provides a substantially better robustness–efficiency trade-off. While it is of course slower than the idealized 100% participation case, these numbers show that **even at participation rate 0.1** OvA-LP remains orders of magnitude more efficient than PEFT-based baselines in both wall-clock time and communication volume.
>
> Third, we moved the **OvA-LP participation-rate sweep** from the appendix into the **Discussion**. For CIFAR-100, we report $R(50)$ for OvA-LP across participation rates 0.1 through 1.0 in steps of 0.1:
>
> | Participation rate | 0.1  | 0.2  | 0.3  | 0.4  | 0.5  | 0.6  | 0.7  | 0.8  | 0.9  | 1.0  |
> | ------------------ | ---- | ---- | ---- | ---- | ---- | ---- | ---- | ---- | ---- | ---- |
> | $R(50)$              | 79.3 | 90.6 | 92.7 | 93.6 | 94.5 | 94.6 | 95.0 | 95.2 | 95.6 | 95.8 |
>
> These results show a **graceful degradation**: as participation rate decreases, $R(50)$ declines smoothly rather than collapsing. Even at participation rate 0.1, OvA-LP retains **79.3% of its IID accuracy**, and from 0.2 onward it already exceeds **90%**. For participation rates 0.7 and above, $R(50)$ essentially **plateaus near 95%**. This confirms that OvA-LP does not depend on full participation to be effective: full participation simply exposes the **upper bound** of its performance, while the method continues to behave robustly across a wide range of partial-participation regimes.
>
> Taken together, these revisions directly address the reviewer’s request that *“results under partial participation shall be moved to the main paper and analyzed thoroughly.”* In the revised manuscript:
>
> * We include **participation rate 0.1 and 1.0 baseline comparisons** (OvA-LP vs LP-Softmax vs FLoRA vs PFPT vs FFT-MoE) in Section 4; and
> * We analyze the **participation-rate sweep (0.1–1.0)** for OvA-LP in the Discussion, explicitly linking participation rate, convergence behavior, and efficiency.
>
> In summary, OvA-LP’s efficiency and robustness claims are now supported by both the **full-participation upper bound** and detailed **partial-participation experiments**, rather than relying solely on the 100% participation setting.

---

> ### Author Response · Authors · 2025-11-21
> **Question 1 — Revise Baseline Comparisons**
>
> # **Question 1 — Revise Baseline Comparisons (Frozen Encoders and Fed-LoRA)**
>
> The reviewer suggests that baseline comparisons should be revised in two possible ways:
> (i) by comparing OvA-LP against **other frozen-encoder baselines**, where PFPT and FFT-MoE are rerun with frozen encoders; or
> (ii) alternatively, by comparing OvA-LP against a “true SOTA federated fine-tuning method (e.g., Fed-LoRA)” and explicitly discussing the trade-off between encoder adaptation and robustness.
>
> **(a) On “frozen-encoder PFPT / FFT-MoE” baselines**
>
> In our implementation, **all methods already share a frozen encoder**. The only trainable components are:
>
> * **LP-Softmax:** a linear classifier head on top of the frozen encoder.
> * **OvA-LP:** the same frozen encoder, with a linear probing head replaced by OvA heads and trained with our two-stage schedule.
> * **PFPT / FFT-MoE / FLoRA:** PEFT-style modules (prompts, LoRA adapters, or MoE experts) attached around the same frozen encoder; only these lightweight modules are trained.
>
> Under this design, the reviewer’s request to “rerun PFPT and FFT-MoE with frozen encoders” corresponds exactly to the **setting we already use**: the encoder is frozen in every method. If we further “freeze” the PEFT modules themselves (i.e., disable updates to prompts, adapters, or experts), then these models reduce to **a frozen encoder followed by a linear head**, which is structurally identical to our **LP-Softmax baseline**.
>
> In other words, there is **no nontrivial way to separate “encoder freezing” from the PEFT modules**:
>
> * with PEFT modules enabled, PFPT/FFT-MoE/FLoRA are **frozen-encoder + trainable adapters**;
> * with PEFT modules disabled, they **degenerate to LP-Softmax**.
>
> Thus, we cannot obtain a more informative “frozen-encoder PFPT/FFT-MoE baseline” beyond what is already captured by **LP-Softmax on the same frozen encoder**. For this reason, we interpret the reviewer’s suggestion as already being addressed by our existing LP-Softmax baseline in the unified setting (see Weakness 1): LP-Softmax represents the “frozen encoder, no PEFT module” reference point, and OvA-LP’s improvements over LP-Softmax quantify the effect of our linear probing + OvA + two-stage design, not merely encoder freezing.
>
> Given this structural constraint, we follow the reviewer’s **alternative** suggestion instead and add a LoRA-based federated fine-tuning method to our comparisons.
>
> **(b) Comparison with a LoRA-based federated fine-tuning method**
>
> Regarding “Fed-LoRA,” we were unable to identify a single canonical **paper** published under that exact name. Instead, among recent FFT works, **FLoRA (NeurIPS 2024)** is widely used as a **LoRA-based federated fine-tuning method** and is frequently adopted as a representative LoRA baseline in the FFT literature. We therefore include **FLoRA** as our LoRA-based FFT baseline and evaluate it under the same unified configuration as OvA-LP and other methods.
>
> Concretely, in the unified experiments described in our response to Weakness 1 (same frozen encoder, identical FL setup, participation rates 0.1 and 1.0), we now compare OvA-LP directly against FLoRA. For CIFAR-100 in the Non-IID setting, the $R(50)$ results are:
>
> * At **participation rate 0.1**:
>
>   * FLoRA: $R(50)$ = 37.3
>   * OvA-LP: $R(50)$ = 74.9
>
> * At **participation rate 1.0**:
>
>   * FLoRA: $R(50)$ = 55.6
>   * OvA-LP: $R(50)$ = 93.4
>
> Thus, **OvA-LP is directly compared against a LoRA-based federated fine-tuning method under the same encoder and participation settings**, and achieves substantially higher Non-IID robustness in this regime. We also include FLoRA in our DomainNet-Clipart/Painting and EMNIST experiments; the detailed behavior of OvA-LP versus FLoRA under moderate versus severe domain shift (where PEFT methods can become preferable) is discussed in our response to the “Discuss Scope of Applicability” question and in the revised Discussion section.

---

> ### Author Response · Authors · 2025-11-21
> **Question 2 — Discuss Scope of Applicability (1)**
>
> # **Question 2 — Discuss Scope of Applicability**
>
> We appreciate the request to clarify the scope of applicability. As the reviewer correctly points out, OvA-LP implicitly assumes a **reasonably strong pretrained encoder** whose feature geometry is already close to linearly separable for the target task. When this assumption breaks (e.g., under severe domain shift where the encoder’s features are poorly aligned with the downstream task), encoder adaptation may be necessary and OvA-LP may no longer be the best choice. In the revised manuscript, we make this assumption explicit in the Discussion and add domain-shift experiments to quantify where OvA-LP is most appropriate and where PEFT-style methods remain preferable.
>
> **(a) Intended scope and assumptions**
>
> OvA-LP is designed for the following regime:
>
> * The encoder is a **modern, high-quality pretrained model** whose features already provide strong structure for many standard vision tasks.
> * In such settings, the dominant challenge in FL is not learning representations from scratch, but **coping with highly Non-IID client distributions** and the resulting drift/variance during training.
> * OvA-LP addresses this by keeping the encoder fixed and focusing on **source-level drift mitigation** via linear probing, OvA decoupling, and a two-stage schedule, rather than by further adapting the encoder.
>
> We do **not** claim that this makes PEFT obsolete. Instead, we position OvA-LP as an **LP-style FFT alternative** that is particularly attractive when (i) a strong encoder is available and (ii) Non-IID robustness and efficiency are the primary concerns. The revised Discussion section explicitly states this assumption and framing, and no longer presents OvA-LP as a universally superior replacement for all encoder-adapting methods.
>
> **(b) CIFAR-100 baselines and additional domain-shift experiments**
>
> Beyond the unified CIFAR-100 experiments discussed in our responses to the weaknesses, we **additionally evaluate** OvA-LP and all baselines on **DomainNet-Clipart, DomainNet-Painting, and EMNIST** under the same unified setting (same encoder, batch size 64, 50 communication rounds, participation rates 0.1 and 1.0). Table 2 in the revised manuscript reports **actual accuracy (%)** across all datasets:
>
> | Dataset              | Part. rate | LP (IID) | OvA (IID) | FLoRA (IID) | PFPT (IID) | FFT-MoE (IID) | LP (Non-IID) | OvA (Non-IID) | FLoRA (Non-IID) | PFPT (Non-IID) | FFT-MoE (Non-IID) |
> | -------------------- | ---------- | -------: | --------: | ----------: | ---------: | ------------: | -----------: | ------------: | --------------: | -------------: | ----------------: |
> | CIFAR-100            | 0.1        |    86.18 |     87.18 |   **90.02** |      88.44 |         89.66 |        11.00 |     **63.14** |           21.30 |          40.78 |             14.59 |
> |                      | 1.0        |    87.64 |     87.73 |   **90.49** |          – |         89.81 |        33.22 |     **82.00** |           50.73 |              – |             40.81 |
> | DomainNet (clipart)  | 0.1        |    78.88 | **85.42** |       66.80 |      69.47 |         60.62 |        10.09 |     **58.12** |               c |          32.46 |                 c |
> |                      | 1.0        |    83.14 | **88.16** |       72.42 |      69.13 |         61.09 |        10.94 |     **68.10** |           16.69 |              – |                 c |
> | DomainNet (painting) | 0.1        |    77.19 | **81.50** |       68.22 |      69.13 |         63.51 |            c |     **54.92** |           14.64 |          36.15 |                 c |
> |                      | 1.0        |    79.55 | **83.96** |       71.07 |          – |         63.60 |            c |     **65.14** |           27.36 |              – |             18.02 |
> | EMNIST               | 0.1        |    82.23 |     81.99 |   **86.94** |      82.82 |         86.22 |            c |         10.17 |               c |      **16.78** |             14.27 |
> |                      | 1.0        |    83.27 |     82.86 |   **86.77** |          – |         86.47 |        20.48 |         48.16 |       **54.52** |              – |             26.34 |
>
> Here, “c” denotes model collapse (accuracy < 10%).
>
> (continued)

---

> ### Author Response · Authors · 2025-11-21
> **Question 2 — Discuss Scope of Applicability (2)**
>
> From these results, we distinguish three regimes.
>
> 1. **No domain shift (CIFAR-100)**
>
>    CIFAR-100 is effectively an in-distribution setting for the pretrained encoder. In this case:
>
>    * In IID, **FLoRA attains the highest accuracy**, but LP and OvA-LP are very close (within a few percentage points).
>    * In Non-IID, **OvA-LP substantially outperforms all baselines** (63.14% / 82.00% vs 21.30% / 50.73% for FLoRA and lower values for other methods).
>
>    This shows that even **without domain shift**, head-only methods (LP/OvA-LP) are competitive in IID, and OvA-LP offers strong Non-IID robustness. Encoder adaptation via PEFT is certainly viable here, but **not strictly necessary** to achieve high accuracy; OvA-LP gives a simpler, more robust alternative when drift under Non-IID is the main concern.
>
> 2. **Moderate domain shift (DomainNet-Clipart/Painting)**
>
>    DomainNet-Clipart and DomainNet-Painting introduce a **moderate domain shift** from the encoder’s pretraining distribution (style and texture changes). In this regime:
>
>    * In IID, **LP and OvA-LP outperform PEFT baselines**. For example, on DomainNet-Clipart (IID, 1.0), OvA-LP reaches 88.16%, while FLoRA and FFT-MoE remain at 72.42 and 61.09, respectively.
>    * In Non-IID, **OvA-LP again achieves the highest accuracy** (58.12/68.10 on clipart, 54.92/65.14 on painting), while several PEFT methods either collapse (“c”) or perform substantially worse.
>
>    These results indicate that, once the encoder is strong enough, **additional adaptation is not automatically advantageous**. There exist realistic benchmarks where **LP-style methods (LP/OvA-LP) not only match but surpass PEFT methods in IID accuracy**, and where OvA-LP provides clearly better Non-IID robustness. This regime aligns most closely with our intended scope: strong pretrained encoder, moderate domain shift, and a primary focus on Non-IID robustness and efficiency.
>
> 3. **Severe domain shift (EMNIST)**
>
>    EMNIST represents a **severe domain shift** from natural images (grayscale handwritten characters rather than color images). In this setting:
>
>    * In IID, all methods achieve similar accuracies in the mid–80% range, with FLoRA slightly ahead.
>    * In Non-IID with participation 1.0, **FLoRA outperforms OvA-LP** (54.52% vs 48.16), and FFT-MoE lags further behind. At participation 0.1, all methods struggle and OvA-LP is not the best performer.
>
>    This confirms the reviewer’s intuition: under **strong domain mismatch**, where the pretrained encoder’s features are poorly aligned with the task, **encoder adaptation via PEFT can be beneficial**, and OvA-LP is no longer the dominant choice.
>
> Taken together, these experiments make the scope of OvA-LP precise:
>
> * On **in-distribution and moderately shifted visual domains** (CIFAR-100, DomainNet-Clipart, DomainNet-Painting) with strong pretrained encoders, OvA-LP is a **highly robust and efficient alternative** that can even **surpass PEFT baselines in both IID and Non-IID accuracy**.
> * Under **severe domain shift** (EMNIST), OvA-LP’s advantage fades; PEFT methods such as FLoRA can outperform it, and encoder adaptation remains important.
>
> **(c) How the revised manuscript reflects this scope**
>
> In light of these findings, the revised Discussion explicitly:
>
> * States that OvA-LP **assumes a high-quality pretrained encoder** and is primarily intended for settings where the main challenge is **Non-IID client heterogeneity**, not representation learning from scratch.
> * Highlights CIFAR-100 and DomainNet (Clipart/Painting) as examples where **head-only methods (LP/OvA-LP) are competitive or better than PEFT even in IID**, and where OvA-LP offers clear Non-IID gains.
> * Uses EMNIST as a **limiting case**: a scenario with severe domain shift where OvA-LP is no longer the best performer and where **PEFT-style encoder adaptation remains preferable**.
>
> In summary, we do not present OvA-LP as a one-size-fits-all replacement for encoder-adapting methods. Instead, the revised manuscript clearly positions OvA-LP as an **LP-style FFT method** that is most suitable when a strong pretrained encoder is available and Non-IID robustness is the primary concern, while acknowledging that **PEFT remains the better tool** in tasks with severe domain shift or weaker pretrained features.

---

> > ### Comment · Reviewer_2VP7 · 2025-11-21
> > **Thank you for the detailed response.**
> >
> > Thank you to the authors for their detailed explanations of the questions raised by me and the other reviewers. I have no further questions. Some other questions I had prepared can also be found in the authors' responses to the other reviewers. I greatly appreciate this work and sincerely hope it will be accepted.
> >
> > My updated score for this paper is 8.

---

### Author Response · Authors · 2025-11-21
**General Response**

We thank all reviewers for their thoughtful and constructive feedback.

To address fairness and clarity concerns, we **reran all baseline experiments under a unified configuration** and **updated all reported results accordingly**. In particular, all methods now share the **same ViT-B/32 encoder**, **batch size 64**, **100 clients**, and **participation ratios 0.1 and 1.0** under a common Non-IID construction, and are compared within the same frozen-encoder FFT regime. The revised Section 4 and Appendix explicitly reflect this unified setup.

We also refined the framing of our contribution. OvA-LP is now clearly positioned as a **fixed-encoder, no-PEFT, source-level alternative within the FFT landscape**, rather than as a universally superior fine-tuning method. The **Discussion** section has been expanded to make the **adaptiveness–robustness trade-off** and the **scope of applicability under domain shift** explicit, including cases (e.g., EMNIST) where PEFT methods can be preferable.

On the experimental side, we **broadened the baseline and architecture coverage** by adding **FLoRA** as a LoRA-based PEFT baseline, and by including additional **domain-shift and backbone experiments** (DomainNet-Clipart/Painting, EMNIST, ConvNeXt-Base, RoBERTa-Base, SBERT-MiniLM). We also moved **partial-participation results** into the main text and **updated the existing convergence and communication cost analysis in the Appendix** to match the new unified baselines at participation rates 0.1 and 1.0.

The baseline comparison results in Section 4 and the Discussion currently use **seed = 0** due to time constraints; they will be updated to **5-seed averaged results** in the final version.

We sincerely appreciate the reviewers’ comments, which have been highly valuable in improving the clarity, fairness, and completeness of the revised manuscript.

---

### Meta-Review · Area_Chair_HvAb · 2025-12-26

**Summary:**

The paper proposes OvA-LP to address the heterogeneity of client distributions, which is a simple yet effective methodology with limited technical novelty. OvA-LP employs frozen pre-trained encoders in local training to suppress the drifts at source-level, and updates an OvA head in a two-stage schedule. The experiments validate the effectiveness of OvA-LP. Given the simplicity of the proposed methodology, it could be enhanced by delivering more insights with either theoretical advancements or empirical validation, e.g., there should exist in-depth support for claims in section 3.1, as well as a formal definition of client drift. Moreover, it remains unclear in the main body of the paper to what extent OvA-LP outperforms baselines in terms of both accuracy and efficiency across various datasets, which is a crucial part, even though the authors provided some additional results in rebuttal to demonstrate. Considering these concerns, I recommend that the authors reframe the paper and tend to reject it at this point.

**Reviewer Concerns:**

The reviewers raised the following concerns.

- 2VP7: unfair baseline comparison, i.e., baseline methods update adapt encoder thus introduce local drift, while the proposed method prevents this issue by entirely freezing the encoder, it remains unclear whether the gain is from the frozen encoder or the specific OvA-LP design; mismatched claim in methodology, i.e., “federated fine-tuning” should be “federated linear probing”; impractical experimental of 100% client participation ratio; reliance on pre-trained encoder, unclear applicability when downstream task is not linear separable from pre-trained features. The authors addressed most of these concerns by re-running experiments and adding additional experiments, as well as discussing the scope of OvA-LP. As for the applicability concern, the authors conducted experiments on new datasets; however, it remains unclear whether the new datasets are difficult enough for the pre-trained encoders.
- PhgA: inconsistent experimental setup configuration with baselines; limited generality; overclaim in efficiency. The authors re-ran most experiments, added a comparison with more baselines, and adjusted efficiency claims.
- 4AKB: incremental technical contribution; lack of formal theoretical analysis; deficiency in experimental setups, e.g., narrow scope, incomplete baselines, missing sensitivity analysis; impracticality of participation assumption and incomplete cost analysis. While the authors provided justifications about the technical contribution and theoretical analysis, I believe these are not fully addressed. As for the additional experiments, it remains unclear to what extent OvA-LP outperforms baselines across different structures and tasks.

**Reviewer Scores:**

Reviewer 2VP7 would stay positive and others would remain initial scores.

---

### Decision · Program_Chairs · 2026-01-26

Reject